# Variability in the location of high frequency oscillations during prolonged intracranial EEG recordings

Stephen V. Gliske[1,2], Zachary T. Irwin[3], Cynthia Chestek[3], Garnett L. Hegeman[2], Benjamin Brinkmann [4], Oren Sagher[5], Hugh J.L. Garton[5], Greg A. Worrell[4] & William C. Stacey [1,3]

The rate of interictal high frequency oscillations (HFOs) is a promising biomarker of the seizure onset zone, though little is known about its consistency over hours to days. Here we test whether the highest HFO-rate channels are consistent across different 10-min segments of EEG during sleep. An automated HFO detector and blind source separation are applied to nearly 3000 total hours of data from 121 subjects, including 12 control subjects without epilepsy. Although interictal HFOs are significantly correlated with the seizure onset zone, the precise localization is consistent in only 22% of patients. The remaining patients either have one intermittent source (16%), different sources varying over time (45%), or insufficient HFOs (17%). Multiple HFO networks are found in patients with both one and multiple seizure foci. These results indicate that robust HFO interpretation requires prolonged analysis in context with other clinical data, rather than isolated review of short data segments.

[1] Department of Neurology, Comprehensive Epilepsy Program, University of Michigan, 1500 E. Medical Center Dr, Ann Arbor, MI 48109, USA. [2] Department of Neurology, Sleep Disorders Center, University of Michigan, 1500 E. Medical Center Dr, Ann Arbor, MI 48109, USA. [3] Department of Biomedical Engineering, Biointerfaces Institute, University of Michigan, 2800 Plymouth Rd., NCRC Bldg. 10, Ann Arbor, MI 48105, USA. [4] Departments of Neurology and Physiology and Biomedical Engineering, Mayo Systems Electrophysiology Laboratory, Mayo Clinic, 200 First St. SW, Rochester, MN 55905, USA. [5] Department of Neurosurgery, University of Michigan, 1500 E. Medical Center Dr, Ann Arbor, MI 48109, USA. Correspondence and requests for materials should be addressed to S.V.G. (email: sgliske@umich.edu) or to W.C.S. (email: william.stacey@umich.edu)

Finding new biomarkers in epilepsy is crucial, as epilepsy is one of the world's most common neurological diseases, and one-third of patients do not respond to medication[1]. One of the primary clinical tools to help patients with intractable epilepsy is epilepsy surgery, which attempts to resect the region of brain responsible for generating seizures, known as the epileptogenic zone. However, there is no perfect method to identify the true epileptogenic zone. Standard clinical practice is to place intracranial electrodes and observe spontaneous seizures, with the patient remaining in the hospital for many days. Clinicians estimate the true epileptogenic zone by determining the seizure onset zone (SOZ), which guides surgical resection. The SOZ is only an estimation of the true epileptogenic zone: it is the best current standard but removing it does not always lead to seizure freedom, motivating the search for additional biomarkers[2,3]. The final resected volume (RV) is not necessarily the same region, as it is limited by anatomical and functional considerations. This procedure has been relatively unchanged in decades. The success rate has also been relatively unchanged, ranging from around 70% seizure freedom in temporal lobe epilepsy to 40% in extratemporal, nonlesional epilepsy[4]. This has led to a search for other diagnostic biomarkers that could help improve or at least predict outcomes[5].

High frequency oscillations (HFOs) are a promising new candidate to identify epileptogenic tissue. They are short, infrequent EEG waveforms with peak frequencies in the 80–500 Hz range[6–11]. There is hope that HFOs may be able to improve localization of the epileptogenic zone, and potentially do so without waiting for seizures to occur by using brief segments of data between seizures, known as the "interictal" period[12]. HFOs were originally described in normal brain activity[13], but a large body of research has now established that they have a strong correlation with epilepsy, both when detected manually[2,14–21] or by computer algorithms[22–28]. Thus, HFOs potentially provide a means for clinicians to identify seizure networks with greater resolution than previous clinical tools[10,11].

A large portion of the current clinical HFO literature is based upon a strategy of analyzing 10 min of slow wave sleep, which allows for experts to process and count HFOs manually[12]. Slow wave sleep was chosen because it has fewer muscle artifacts and the rate of HFOs is higher. This strategy has led to a great deal of clinical data showing the strong relationship between epileptic tissue and HFO rate[12]. However, this strategy was based primarily upon practical implications: manual processing is prohibitively time-intensive, so manual processing is not performed on larger segments[20]. Basing a decision upon such a small amount of data inherently leads to the question of whether 10 min is sufficient to characterize the epileptogenic network fully—especially since it is being compared to a traditional analysis that often utilized up to 2 weeks of continuous ictal and interictal EEG. One prior work with human-scored HFOs suggested that results were similar across multiple days when only 1 min of data was scored each night[29]. Similarly, other work with short datasets has evaluated whether the region of highest HFO rate changes with respect to sleep stage, proximity of a seizure, and withdrawal of medications[29–31] and found that while the rate may change, the predicted region does not. However, these studies were typically limited to brief datasets in a modest number of patients and no control group without epilepsy. One exception is a comparison of HFO rate versus sleep stage[25]. In that study, an automated HFO detector was used in cohort of 15 epileptic patients, and HFO rates on two consecutive nights were found to vary over time due to sleep stage and location within the brain. Variability within the same sleep stage was noted but not characterized. To date, no publications have tested the consistency of HFO rates within interictal, slow wave sleep, using the full set of data over the full recording period.

We hypothesize that HFO rates are not stable over the full record: that different 10-min segments of data may have different groups of channels with high HFO rates, even when controlling for sleep stage. The rationale of this hypothesis is the common clinical observation that one day of recording, or even one recorded seizure, is usually insufficient to characterize a patient's seizure network. Clinical experience has shown that some patients have seizures arising from more than one location, and sometimes the different foci are active at different times[32]. HFOs, as a biomarker of the seizure foci, may likewise show similar variability.

A primary limitation of previous HFO work has been the significant time necessary to process the data by hand. One method to analyze larger datasets is to use automated HFO detectors, which have evaluated nearly two weeks of data in some patients[26]. Many HFO detection algorithms have been proposed to increase throughput[9,24,33–35] but often suffer from many false positives because common EEG artifacts may induce "false" HFOs[36]. With great care to avoid artifacts, it is possible to use automated detectors to rapidly analyze vast amounts of data, and begin to explore the characteristics of HFO waveforms[27,37–39]. We used a fully automated "quality HFO" (qHFO) algorithm[26] designed specifically for continuous EEG, which ignores any detection occurring during an artifact or period of poor data quality. The algorithm was previously validated to yield similar results as a human reviewer, and allows us to analyze HFO features in vast datasets while avoiding interrater variability.

This paper utilizes data from 121 subjects, including 18 with full recordings (1–12 days) and 12 control patients without epilepsy implanted with intracranial electrodes for therapeutic electrical stimulation for drug resistant facial pain. The 18 patients with full recordings also had sleep scoring to identify all periods of non-rapid eye movement (NREM) sleep, which allows us to compare directly with previous work using data from only NREM sleep[12,30,31]. We first assess whether elevated HFO rates are correlated with the SOZ, and whether that association is similar when using full records versus 10-min segments. We observed four general categories of variability/stability and developed an automated method to categorize subjects' HFO rate data into these categories. This article describes the algorithm and presents the results of the categorization for all 121 subjects, providing a large-population assessment of spatiotemporal variability in HFO rates, i.e., how HFO rates change over both space and time. These results provide essential information for developing clinical protocols which utilize HFOs for resective surgery planning. We find that HFO rates are indeed predictive of the epileptogenic zone, but are often not consistent when comparing different 10-min epochs. Furthermore, HFOs are found to have temporal variability that, in some subjects, corroborates with temporal changes in other clinical data. We therefore recommend that HFO detections be reported as a function of time throughout the invasive monitoring period and be included as adjunctive data rather than being used for diagnosis in isolation of other clinical data.

## Results

**Association between interictal HFO rates and SOZ.** We first tested the hypothesis that HFO rates are increased within the clinically determined SOZ using prior methods (10 min of slow wave sleep), as well as using all data from the entire hospitalization. We restricted analysis to HFOs detected during sleep in order to compare directly with past work. For the Mayo ($N = 91$) and Control ($N = 12$) cohorts, we had two hours of

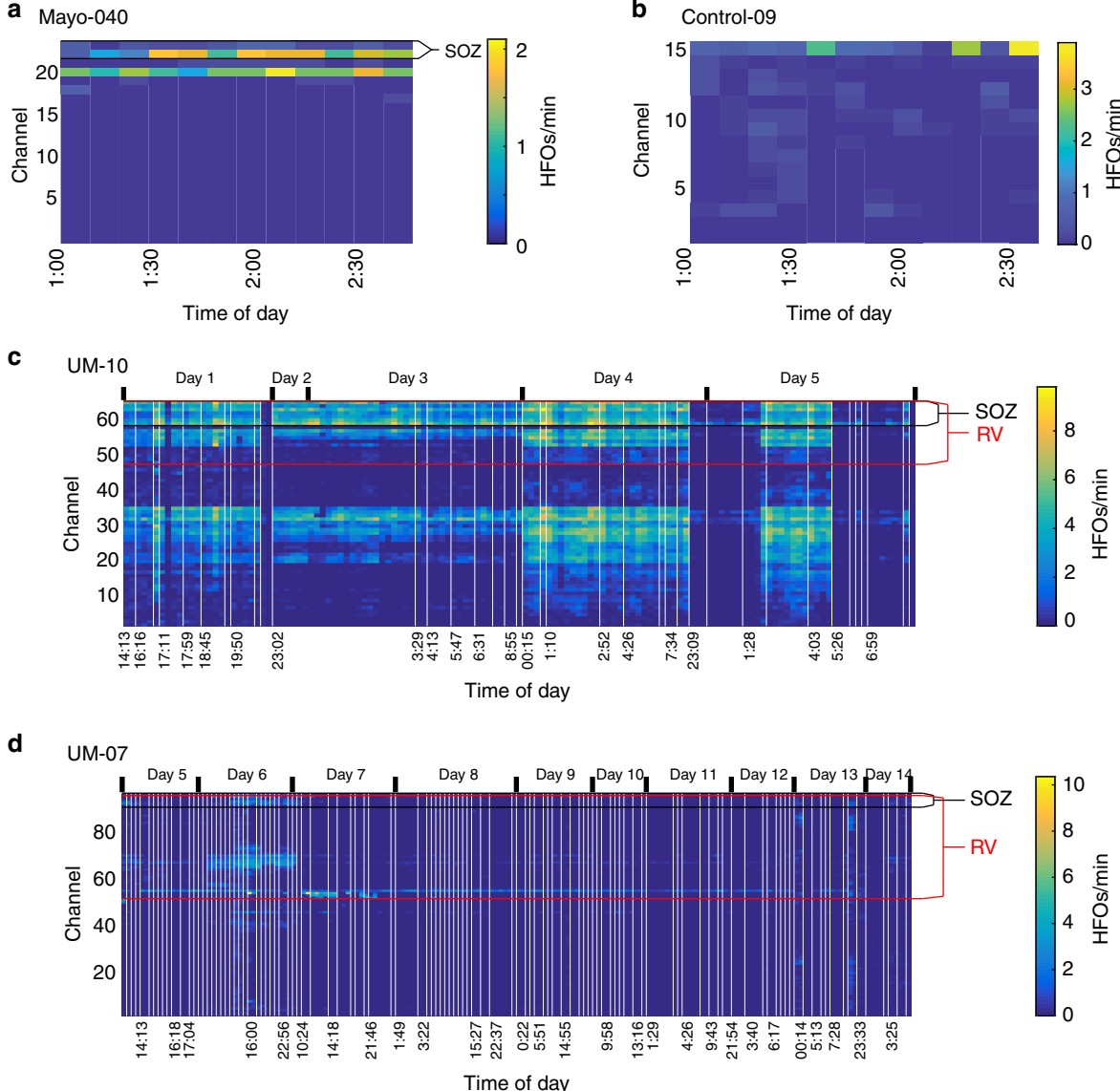

**Fig. 1** Example rates of HFOs. The HFO rate per epoch of NREM sleep is shown for four example patients. All NREM data are concatenated together, with white vertical lines indicating breaks between NREM periods. **a** A small group of channels have high HFO rates, though the highest channel varies over time. **b** One channel is high but alternates with arbitrary groups with much lower rate. **c** Many channels are high in an organized cluster and all appear well correlated. **d** Much more complex patterns are also observed, in which different channel groups dominate on different days. Channels in the resected volume (RV) and clinically determined seizure onset zone (SOZ) are ordered together, as are channels with similar temporal dynamics in HFO rates. Note that patients in **a** and **b** did not have a resection

interictal data recorded from 1–3 AM for each patient. This epoch was chosen in order to maximize the likelihood of NREM, as there were no scalp recordings to allow determination of sleep stage. The University of Michigan (UM) cohort (N = 18) had multiple days and sleep scoring, which allowed inclusion of all available NREM, interictal data, collected from more than a week of recording in many patients. Note, the UM cohort was analyzed in two ways: 1) using all the expert-verified NREM sleep data or 2) using just 1–3 AM data (i.e., without sleep stage verification) for direct comparisons with the Mayo data. HFOs were detected using the automated qHFO algorithm[26], as well as two other automated detection schemes, as HFO detection would be infeasible by hand. These data allowed us to observe how the rates in each channel changed over time. Figure 1 shows HFO rates from example patients from each cohort for the qHFO algorithm.

Our detector allows for analysis of the entire record, which includes a great deal more data than past work using manually scored 10-min segments. The purpose of this first analysis is to determine whether the results from the shorter segments are similar to those from the whole record. We use HFO rate asymmetry (i.e., inside versus outside the SOZ) as a nonspecific measure of how well HFO rate is correlated with the SOZ. We note that the asymmetry provides evidence that the HFOs are clinically relevant, but is retrospective and cannot be used for prospective determination of SOZ. A perfect asymmetry of 1 implies all HFOs are in the SOZ, −1 implies all HFOs are outside the SOZ, and 0 implies that the mean HFO rate is equal inside and outside the SOZ. Thus, the purpose of this test is two-fold: to evaluate whether the detected HFOs are correlated with SOZ (i.e., have an asymmetry close to 1); and to compare those results using either individual 10-min segments or all 10-min segments

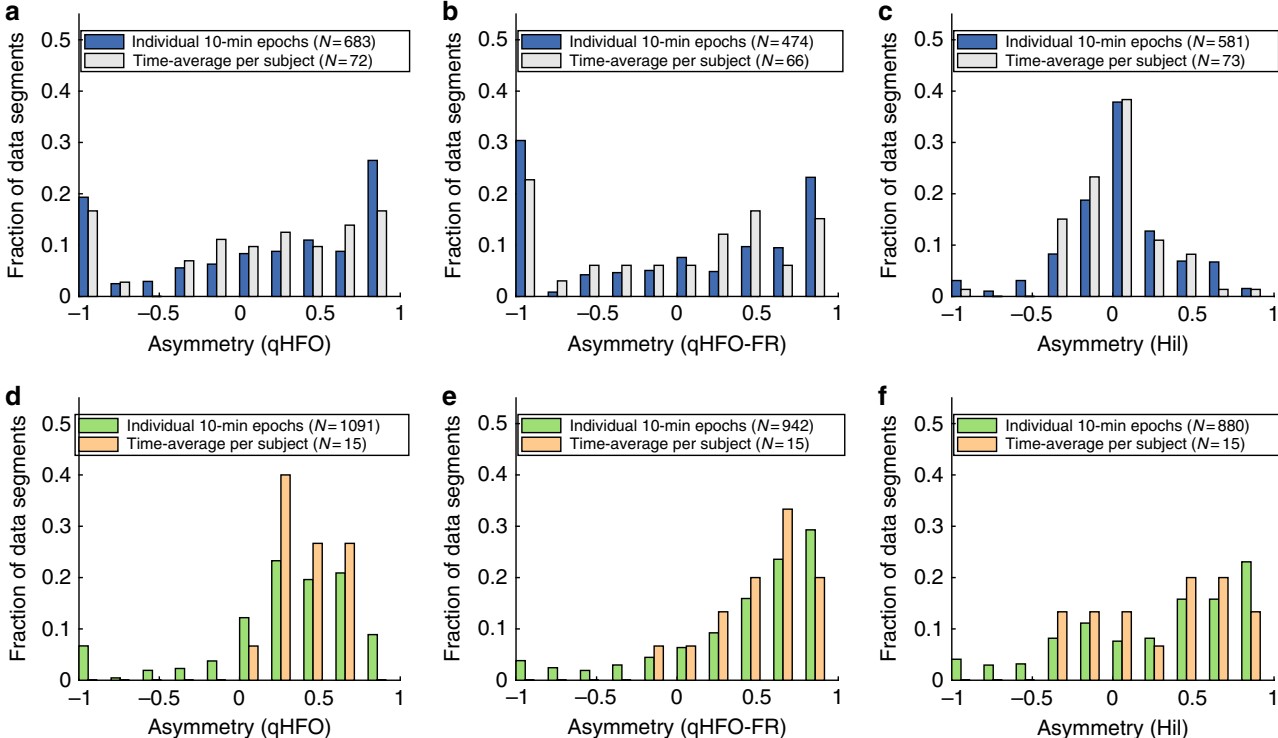

**Fig. 2** Distribution of HFO rate asymmetries. Results are shown for both the Mayo cohort (**a–c**) and the UM NREM cohort (**d–f**), and for three different HFO detectors: qHFO (**a, d**), fast ripple enhanced qHFO (qHFO-FR) (**b, e**), and a Hilbert transform based detector (**c, f**). Asymmetry is the difference between average HFO rate inside versus outside the SOZ, divided by the sum. Asymmetries for HFO rates are displayed in histogram form for all available 10-min epochs in the cohort (blue, green) and averaged over all 10-min epochs in each individual patient (gray, orange). The asymmetry was only computed for cases with at least one channel having 0.5 HFOs/min. HFOs were determined to be related to SOZ if the median rate was significantly positive (right-tailed Wilcoxon Sign-Rank test, $p < 0.05$), seen as a skew to the right in the plots. HFOs from all three detectors in the UM cohort, and from the qHFO detector in the Mayo cohort, were associated with SOZ. There was no significant difference between using just 10 min of data or all data together in any cohort or for any detector (Wilcoxon Rank-Sum test, $0.1 < p < 0.9$). Note, several patients had an asymmetry of −1 when there were no HFOs in the SOZ and at least one channel having >0.5 HFOs per min outside the SOZ. This often occurs in data with low signal quality, and thus occurs much more frequently in patients with worse signal to noise ratios

together. Note that in the Mayo and Control cohorts there were 12 continuous segments, and in the UM cohort there was an arbitrary number based upon all data from the entire hospitalization. The results are displayed as a histogram of all individual 10-min segments ("Individual 10-min epochs") and the histogram of results from each of the patient's whole NREM sleep or 1–3 AM record ("Time-average per Subject"). Note that the Mayo data were not proven to be in NREM sleep during this period. For a comparison, we also computed results from UM using just the first 1–3 AM segment, and the results were indistinguishable from the sleep-scored data (not shown). Patients in whom there were no channels present with a rate >0.5 HFOs/min were excluded from this analysis, as this rate is unlikely to represent epileptic activity.

The clear skew towards "1" seen in Fig. 2a,d, indicates HFOs are correlated with the SOZ. Indeed, the median asymmetry was positive in all four of these distributions, three of which were significant with $p < 0.0001$ (right-tailed Wilcoxon signed rank test). The Mayo "Time-average per Subject" distribution had $p = 0.06$. Additionally, no appreciable difference is observed between the two distributions ("Individual 10-min epochs" and "Time-average per Subject") within a given cohort ($p = 0.5$, Mayo, $p = 0.6$ UM NREM, Wilcoxon Rank-Sum). Note that it is not possible to compare results between the UM and Mayo cohorts as they contain very different numbers of patients and duration of data. We then verified that these results were not dependent upon the qHFO algorithm by repeating the analysis

using fast ripples only (qHFO-FR: Fig. 2b,e), as well as a completely different HFO detector (Hil: Fig. 2c,f). Again, there was no statistical difference in the medians between the 10-min segments and the entire sleep record ($0.1 < p < 0.9$, Wilcoxon Rank-Sum). However, while the median asymmetries are all positive, they do not all reach statistical significance (right-tailed Wilcoxon Sign-Rank test). Specifically, the qHFO-FR was significant in both cases for the UM cohort ($p < 0.0001$) but not Mayo ($p > 0.4$). Although the median was very close to zero in the Mayo data, the Hilbert-based detector was significant for all four groups ($p < 0.05$ in Mayo, $p < 0.01$ in UM). Note that the differences in the Mayo cohort are somewhat difficult to judge because that group had a high number of patients who did not benefit from surgery (see Table 1). The UM data, which had more patients with good outcomes and full sleep scoring, shows the relationship much more clearly. However, when stratified by patient outcome there was no clear relationship with asymmetry: results for patients with Class I outcome (Supplementary Fig. 1) were similar to those with Class II–V outcome (Supplementary Fig. 2) in both cohorts. The results were also similar when comparing the resected volumes (Supplementary Figs. 3 and 4). Thus, asymmetry is not capable of identifying which patients had a good outcome: it is a nonspecific, retrospective measurement that compares regions rather than specific channels. Nevertheless, these results serve as a validation of HFOs as a potential biomarker. While some detectors have diminished performance in some cohorts, the HFOs are associated with epileptogenic

**Table 1 Patient and center data**

**A**

| Cohort | Total number | Adults (age > 18.0) | | | Children | | |
|---|---|---|---|---|---|---|---|
| | | **Number** | **Age** | **Percent female** | **Number** | **Age** | **Percent female** |
| Mayo | 91 | 78 | 35.5 ± 12.2 (19–70) | 62% | 13 | 11.7 ± 5.2 (1–18) | 62% |
| Control | 12 | 12 | 54.7 ± 11.5 (44–68) | 83% | 0 | NA | NA |
| UM | 10 | 8 | 32.6 ± 7.0 (24–43) | 50% | 2 | 8.0 ± 1.4 (7–9) | 50% |

**B**

| | Class I | II | III | IV | V | No resection |
|---|---|---|---|---|---|---|
| Mayo | 28 | 6 | 11 | 11 | 24 | 11 |
| UM | 10 | 4 | 0 | 0 | 0 | 4 |

**C**

| | qHFO | qHFO-FR | HIL | Sum |
|---|---|---|---|---|
| UM | 619,824 | 349,776 | 556,355 | 1,525,955 |
| Mayo | 119,186 | 89,919 | 300,199 | 509,304 |

A: Demographic data are based on available clinical reports. Age is at time of electrode implantation, and was not available for all patients. Age is listed as mean plus or minus the standard deviation, with the minimum and maximum values listed in parentheses
B: The patient ILAE outcome class from each center shows that the Mayo cohort had many patients with poor outcome, so it is difficult to compare HFO results between the two centers
C: The number of detected HFOs from each center using each of the three detection schemes is shown. Data from UM include all NREM sleep during the whole hospitalization, and from Mayo are continuous data from 1–3 AM

tissue, regardless of detector. Furthermore, all HFO detectors show that the association of HFO rate with SOZ is statistically similar whether using just a 10-min segments or longer segments to compute the HFO rate.

**Categorization of temporal variability in HFO rates**. The asymmetry analysis is not sensitive to either specific electrode location or consistency over time, as it averages over time and over all channels within and without the SOZ. We first tested whether the location of the peak HFO rate changes over time. We determined whether the channel with maximum HFOs was stable over time, and how often the "maximum channel" was within the SOZ and resected volume. We found that the location of the maximum channel varies greatly when more than one 10-min epoch is tested (Supplementary Fig. 5). These results are similar to previous work showing that simply choosing the channel with maximum HFOs is not always reliable for identifying the SOZ[26]. This led to our next hypothesis, that the location where HFOs are highest is not consistent over the course of the whole hospitalization. To test this hypothesis rigorously on a full spatiotemporal scale, another approach is needed which includes both 1) a method to quantify variability, and 2) a method to group channels based on the temporal distribution of HFO rates. In preparation for the first step, we visually analyzed the HFO rates per patient and found several typical patterns. Some patients have HFOs restricted to a small number of electrodes, although the dominant channel (i.e., with the highest absolute HFO rate) sometimes changes over time (Fig. 1a). Others have a clear dominant channel interspersed with channels that seem to vary arbitrarily (Fig. 1b), or have HFOs spread over a single broad distribution that tends to wax and wane over time (Fig. 1c). Other patients show yet more complex variability, such as in Fig. 1d, where there are different clusters of active channels at different times. In that specific case, some channels have high HFO rates at all times while other channels have high HFO rates at localized times (such as one group on day 6, another on day 7, and a more widespread group on day 13). After having observed patterns of temporal variability in many patients, we defined four general categories of spatiotemporal variability, and sought to classify each patient into one of these categories: a) a single source that

was consistent over time, b) a single source that varied over time, c) multiple sources varying over time, and d) not enough HFOs present to classify. We can thus prove our hypothesis by showing that "not all patients are category (a)."

The second step, to group channels with similar HFO behavior, was accomplished using blind source separation to determine distinct groups of channels with similar temporal evolution in their HFO rates, as well as quantify the temporal evolution per group. In order to give full clarity to this procedure, we show intermediate steps for one example patient per cohort (Figs. 3–5), using the HFO detections from the standard 80–500 Hz qHFO detector. Afterwards, we show the main results, quantifying the prevalence of each type of variability in all 121 patients (Fig. 6) for all three HFO detection methods.

Example results of the blind source separation for the same patients as Fig. 1 are shown in Fig. 3. The weights of each channel's membership in each group is given by matrix $W$, and the mean HFO rate per the group per epoch of time is given in matrix $H$. In the case where a single group of channels all had similar temporal variability (Fig. 1a,c), the group of channels can be described by one source (Fig. 3a,c). The other examples show more complex behavior. Figure 1a had two dominant channels that alternated over time. In the more complex case (the subject shown in Fig. 1d and Fig. 3d), the channel groups still map to the observed features: channel group 1 accounts for the diffuse activity on night 6, group 2 accounts for the focal activity on day 7, and group 3 accounts for the channel that had consistently increased HFO rates. Figure 3b, which has data from one of the patients without epilepsy, shows a more arbitrary pattern with four different groups alternating over time.

Further insight and confidence in the performance of blind source separation can be obtained by comparing the groups of channels with clinical data such as the clinically determined SOZ obtained from visual review of the standard intracranial EEG during seizures, the resected volume, and the physical locations of the channels (Fig. 4). The four patients from the previous figures are given as examples of the different patterns. Patient Mayo-040 had one source including two channels, but only one of the channels was in the SOZ (Fig. 4a). Control-09 had multiple, varying locations with high HFOs, and although two of the

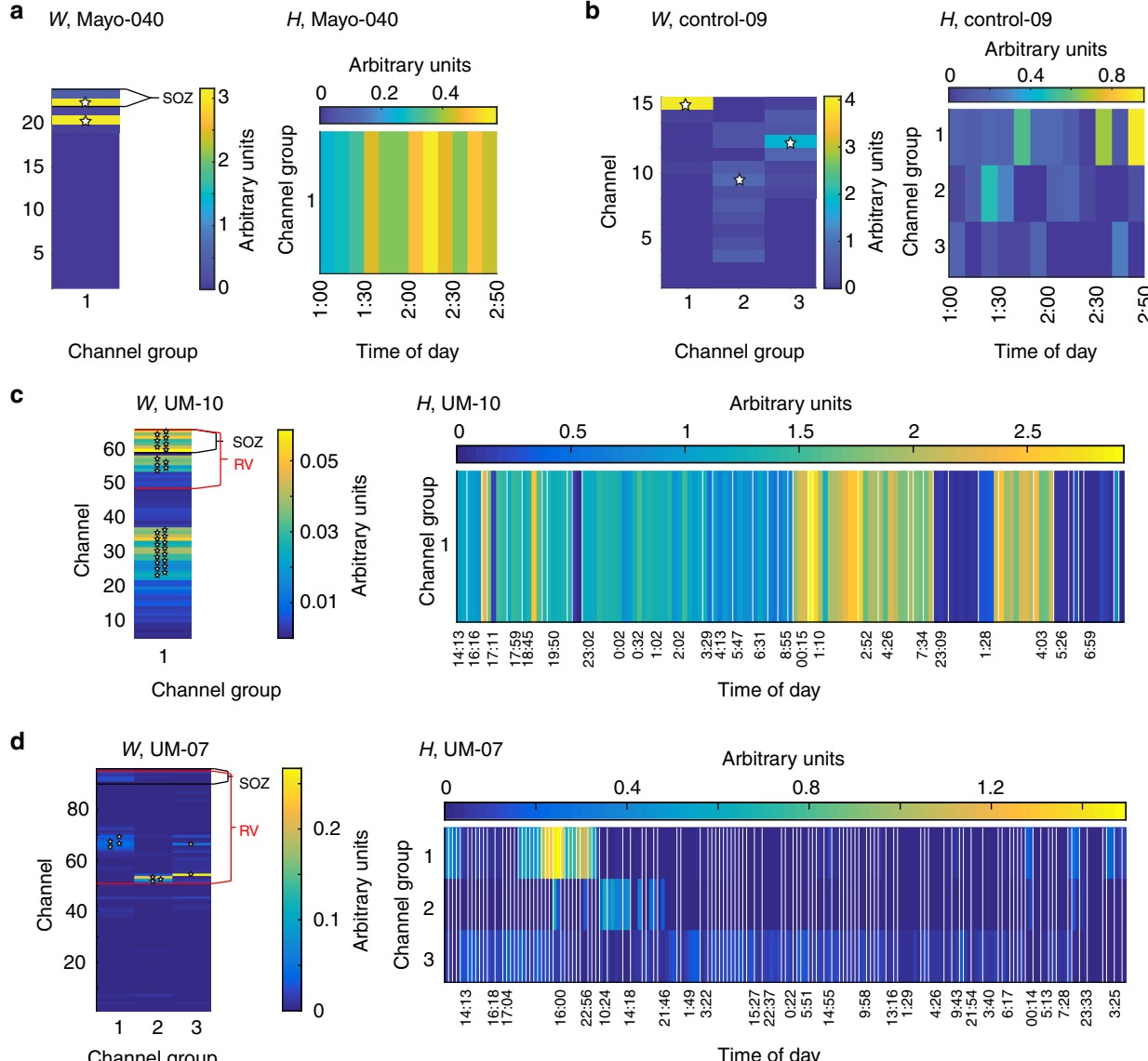

**Fig. 3** Blind source separation of HFO rates. Each of the HFO rates presented in Fig. 1 are decomposed into groups of channels. The associations of each channel with each group (**a–d**, left), denoted by *W*, and each time epoch (**a–d**, right), denoted by *H*, are shown for each example patient. Denoting the rates in Fig. 1 by the matrix *R*, the blind source separation satisfies the matrix product $R = W \times H$. White stars indicate the channels most involved with each source, which are color coded in Fig. 4

locations are juxtaposed, the patient does not even have epilepsy (Fig. 4b). UM-10 has HFOs spread over the majority of the grid, much of which did not appear to be related to the SOZ or resected volume (Fig. 4c). However, the *W* values in Fig. 3c1 demonstrate that the channels with very highest HFO rates were within the SOZ.

UM-07 (Fig. 4d) is quite different, both clinically and in terms of HFO rates. In this patient, 170 subclinical electrographic seizures were recorded over 14 days, with five different ictal onset semiologies (note, types 2 and 3 had similar ictal onset location, but distinct propagation patterns). These seizure onset types did not occur completely randomly but rather seem to show some evolution from one type to the next (Fig. 5): type 1 occurred only days 1, 6 and 14; whereas type 2 and 3 occurred on days 6–7, type 4 on days 7–8, and type 5 on days 8–14. Only three of this patient's typical clinical seizures occurred, all matching type 1.

We also note that the HFO channel groups for patient UM-07 not only show spatial relation with a subset of the seizure types

(Fig. 4d), but also show temporal associations (Fig. 5). For example, channel group 1 (spatially associated with seizure type 4), only has high HFO rates from about 12 h before until just after the time of the first seizure of type 4, about 1–2 AM on day 7 in Fig. 5. In contrast, channel groups 2 and 3 (spatially associated with seizure types 2 and 3) are only high on the days where seizure types 2 and 3 are occurring. No HFO channel groups were spatially associated with seizure types 1 or 5, and very few HFOs occur during the times where seizures of these types occurred. Thus, in this patient it appears that although the HFOs vary considerably over time, they are in fact still intricately linked with seizures, which are themselves also varying considerably over time.

The clinical outcomes for these four patients demonstrate the complexity of these findings. Mayo-040 and Control-09 did not have resections. UM-07 had two resections, one of which had no HFOs, and UM-10 had a resection that removed just the highest portion of the HFO network. Both of those UM patients have had class I outcomes form >3 years.

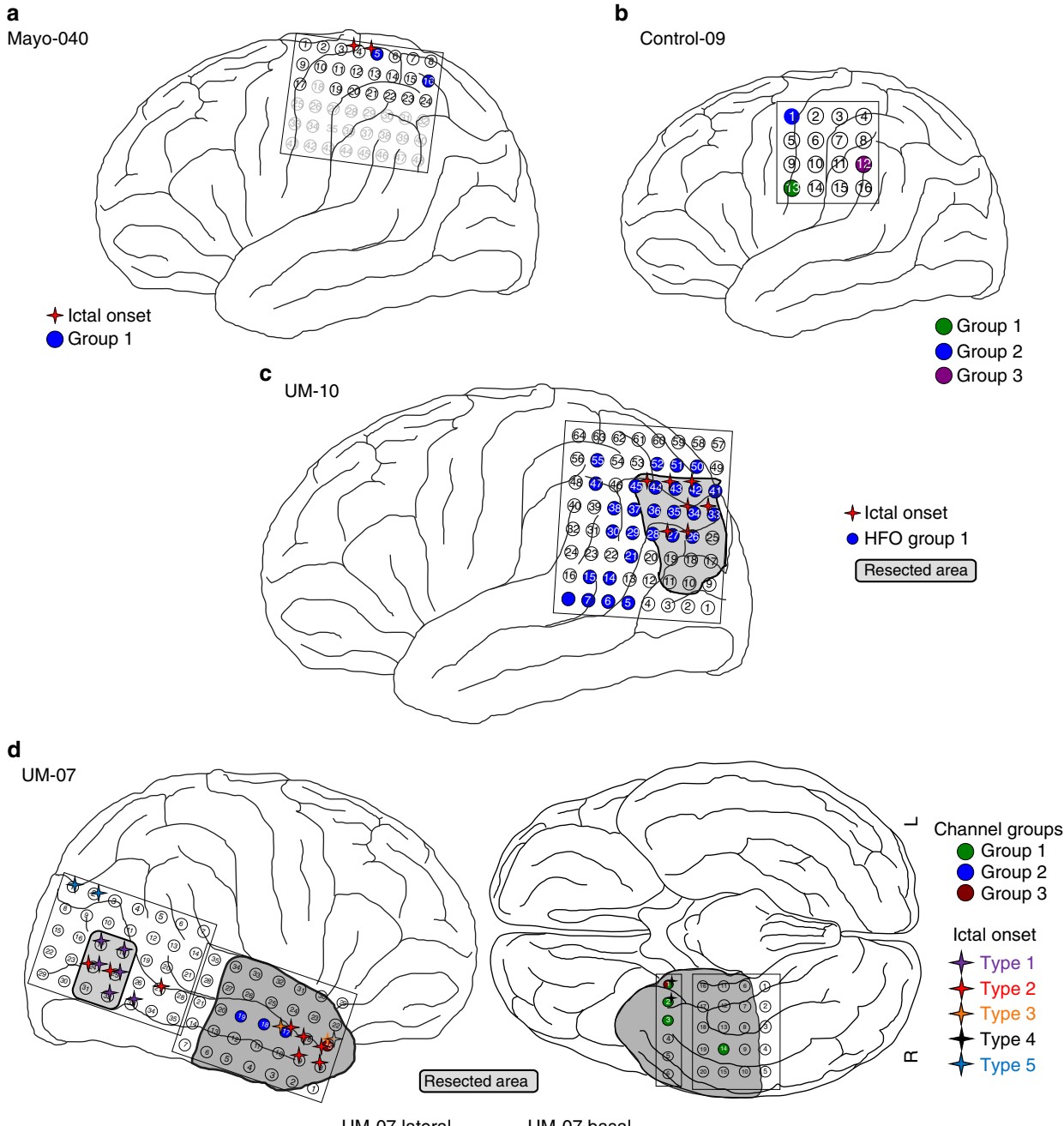

**Fig. 4** Comparison of HFO channel groups and brain regions. Diagrams are shown for the same example patients as in Figs 1 and 3. Electrodes associated with specific channel groups (white stars in Fig. 3) are color coded by group number, and seizure onset electrodes are denoted by colored stars adjacent to the electrode. In **c**, a lesion (not shown) was found under channels 27, 28, 35, and 36. Note, the patients in **a** and **b** did not have resections, but both **c** and **d** did and have had class I outcomes for >3 years. Data were not available for grayed out electrodes in **a**

**Occurrence of each category of spatiotemporal variability.** Our main hypothesis is tested by considering the prevalence of each category of variability for each HFO detection method (Fig. 6). As some subjects in the UM cohort had multiple nights of 1–3 AM data, each night was weighted by the inverse of the number of nights, such that the sum of weights for a given patient was unity. Categorizations were verified by manual review of HFOs on a subset of data for eight of the patients, as manual review of the entire records is infeasible. In all 8 cases, manual review concurred with the automated classification; most importantly, it verified that the three patients classified as category (c) did indeed have different channels active at different times.

In the full 109 subjects with epilepsy (combining Mayo and UM NREM cohorts) with the qHFO detection algorithm, only $(21 \pm 4)\%$ had a single region with consistently high HFO rates, $(15 \pm 4)\%$ had too few HFOs to analyze, and thus $(64 \pm 8)\%$ showed considerable temporal variability. The other detectors likewise found similar results (Fig. 6). In all cohorts and for all detectors, category (a) (a single HFO source consistent over time) never accounted for all the patients with enough HFOs to analyze (i.e., all non-category (d) patients) ($p < 0.05$, $\chi^2$ test). In fact, category (a) never even accounted for the majority of patients for any cohort or detector combination. In the subjects with epilepsy (the Mayo and UM NREM cohorts), we can reject the

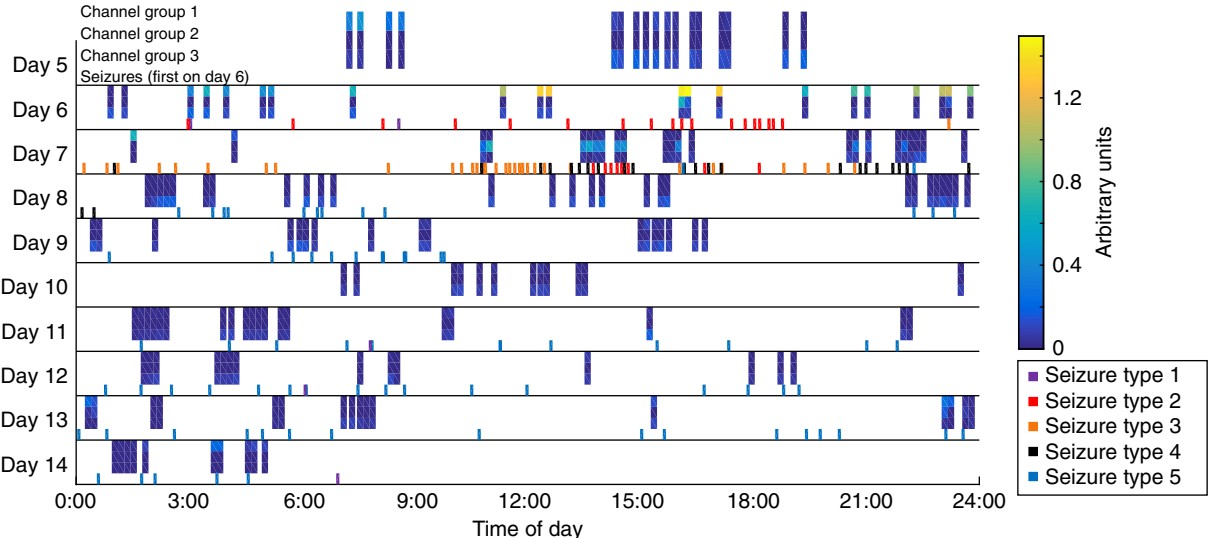

**Fig. 5** Time of seizures versus HFO rates. The effective HFO rates per channel group for patient UM-07 during interictal NREM periods (using the same scheme shown in Fig. 3d) are plotted at the time of day of each epoch to indicate relative activity of each group for a given time. Tricolor vertical bars indicate the three channel groups. The starting time of each seizure (including subclinical seizures) is also denoted as short vertical lines at the bottom of each row, color coded for seizure type. Colorbar: relative strength of each HFO channel group. Note that different HFO groups become active prior to changes in seizure type

null-hypothesis that all patients that had enough data to analyze (i.e., not category (d)) are category (a) with a $p < 10^{-11}$ ($\chi^2$ test).

Little difference was observed between interictal 1–3 AM data versus interictal NREM sleep data in the UM cohort. Some qualitative differences are observed across detectors and cohorts, but none of these are statistically significant (binary $\chi^2$ test between any two cohorts for a given category and detector, $p$-values > 0.05 in all cases). In the UM NREM cohort, where more metadata were available, we also compared the categorizations with age, gender, pathology, SOZ location, and duration of recording. No statistically significant associations were found using any of the HFO detectors ($p > 0.05$, Kruskal-Wallis and Fisher's Exact Test). Of note, there were five patients who were diagnosed during the monitoring with having multiple or diffuse sites of seizure onset. This group would logically be more likely to be category (c), but surprisingly not all of them were, with results that varied slightly with the different detectors (Table 2). It should be noted that in each case the multiple sources were only discovered after the monitoring was performed; thus, restricting the analysis to a single 10-min epoch of HFOs would not have discovered the complex networks. These patients with distributed seizure networks thus pose a special challenge when evaluating HFOs, just as they pose a challenge for traditional EEG monitoring.

**Required amount of recording time**. Given the observed temporal variability, we consider whether a recommended amount of recording time can be determined to allow robust interpretation of HFOs. One way of assessing how much recording time might be needed is to consider whether the categorization using 1–3 AM data is the same over different nights for the same patient. In the UM cohort, 16/18 subjects had more than one nights of sleep. Of these, four had the same category over all nights, four had two categories, and eight had three categories. Thus it was very common (12/16 subjects) for the "answer" to change from one night to the next, i.e., even if a patient had a single consistent source on one night, this answer may not be maintained on different nights.

We next consider a second method to assess how much recording time might be needed for a stable, complete interpretation of HFO rates. For this analysis, we focused on the standard qHFO detections and on the UM cohort, which had markings of interictal NREM sleep and recordings from more than 24 h. We considered the clinical scenario of recording up to a given hour, and then stopping the recording to make a clinical decision about the epileptogenic zone. To mimic this scenario, we used all NREM interictal data recorded up to a given hour to predict the category of HFO variation, scanned over all possible stopping hours, and report category for each stopping hour. Results are shown in Fig. 7 for the qHFO detector. Note that starting with UMHS-0018 all data were acquired through the clinical hardware (Natus), reducing the delays to starting high resolution recording; however, patients tended not to reach NREM sleep before day 2. Results for the other detectors (data not shown) were qualitatively similar. Note, this method of scanning when to stop is similar to the clinical procedure, where all data up to the time of discussion are used to determine the location of the overall seizure onset zone, and often the recording is lengthened because more data are needed for a stable answer.

Whereas some patients stabilized after sufficient data were recorded, other patients continued to have changes. For example, patient UM-07 (patient D in Figs. 1,3,4), stabilizes to category (b) (one channel group that waxes and wanes) after about 3.5 days, and the next five days of recording does not change that category. However, after that period (after nine days of recording and 14 days after the initial implantation surgery), new data were acquired which changed the category to (c): multiple channel groups. Note, the change in the number of channel groups on day 14 precedes a change in ictal onset—later on day 14, a seizure with onset type 1 occurred (the first type 1 seizure since day 6, which was also the last day multiple groups were identified). This seizure on day 14 was the first which matched the patient's usual seizures in over a week, and thus a decision on what area to resect was finally determined on day 14. In other words, this patient actually required all 14 days of recording to identify the proper resected volume, both in terms of seizures and HFOs. In addition, the full extent of the resection in this case extended beyond the

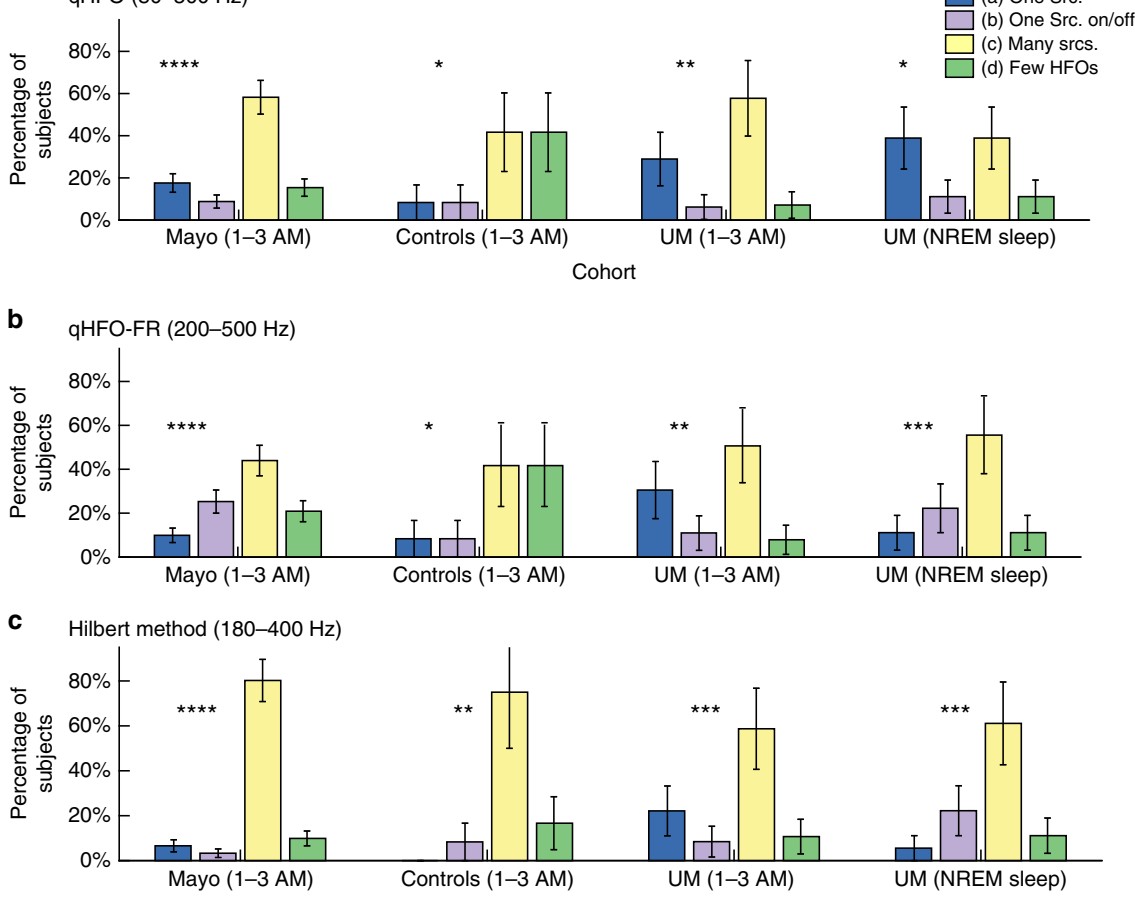

**Fig. 6** Occurrence rates for each category of spatiotemporal variability. Occurrence rates are presented separately for each cohort, separating the results from the NREM interictal data and 1–3 AM data for the UM cohort. **a** HFOs detected using standard 80–500 Hz bandpass. **b** HFOs detected using 200–500 Hz bandpass for Fast Ripples (FR). **c** Hilbert-based detector. No statistical difference was found in the percentage of subjects in a given category between any two cohorts (all $p > 0.05$, $\chi^2$ test). However, in all cases, the category (a) did not account for all patients with measurable HFOs (non-category (d)) ($\chi^2$ test, $*p < 0.05$, $**p < 0.01$, $***p < 0.001$, $****p < 0.0001$). Error bars: 1 s.d

**Table 2 Classifications of multifocal patients**

|         | Class a: one source | Class b: one source on/off | Class c: many sources | Class d: insuff. HFOs |
|---------|---------------------|----------------------------|-----------------------|-----------------------|
| qHFO    | 40%                 | 0%                         | 60%                   | 0%                    |
| qHFO-FR | 0%                  | 40%                        | 60%                   | 0%                    |
| HIL     | 40%                 | 20%                        | 40%                   | 0%                    |

Results are shown for each of the 5 UM patients that had multiple SOZ identified during monitoring, stratified by the three HFO detection methods. Classes correspond to those in Fig. 6

HFO markings—the parietal resection was based upon the patient's typical seizures and a radiographic lesion (that had no HFOs) and the temporal lobe resection was based upon frequent subclinical seizures (and was corroborated by the HFOs). Thus, in this case HFOs would have been helpful but insufficient to identify the full region of epileptic tissue. However, it is also critical to point out that this does not mean the HFOs failed in this patient: they clearly were biomarkers of the epileptic activity, even preceding seizures on several days, but they were insufficient on their own to identify the full epileptogenic zone.

Other patients also had significant changes in HFO rate over the recording session (data not shown). For example, in subject UM-05 the NREM HFO rate dropped significantly during during the last five days of recording. This subject also had no seizures during the 14-day hospital stay and thus determination of SOZ or resection could not be performed. UM-08 required 4 days of recording to reach a stable answer (day 8 post-implant), although was explanted shortly thereafter so it is unclear whether she had fully stabilized. Several patients initially were category (a) then switched to categories (b) or (c). The high variability in the cohort of 18 patients with multiple days and NREM sleep scoring suggests that it is common for patients to have variable HFO rates and locations over the course of the hospitalization.

## Discussion

Many prior clinical reports have shown strong correlations of HFO rate to the seizure onset zone, but there has never been a robust demonstration of whether that answer is stable over time, or how much data are needed to obtain a stable result. By utilizing a validated, automated detection algorithm that redacts artifacts, we were able to analyze HFO rates over long periods of time and multiple days in 121 patients, including control patients without epilepsy. We compared both 1–3 AM data (as putative sleep data), as well as human-scored NREM sleep to avoid any potential confound of changing sleep stages[12,30,31]. We find that many patients have multiple independent regions of high HFO rates. Based on this, we conclude that 10-min of data cannot be construed as representative of the HFO rates at other times in all

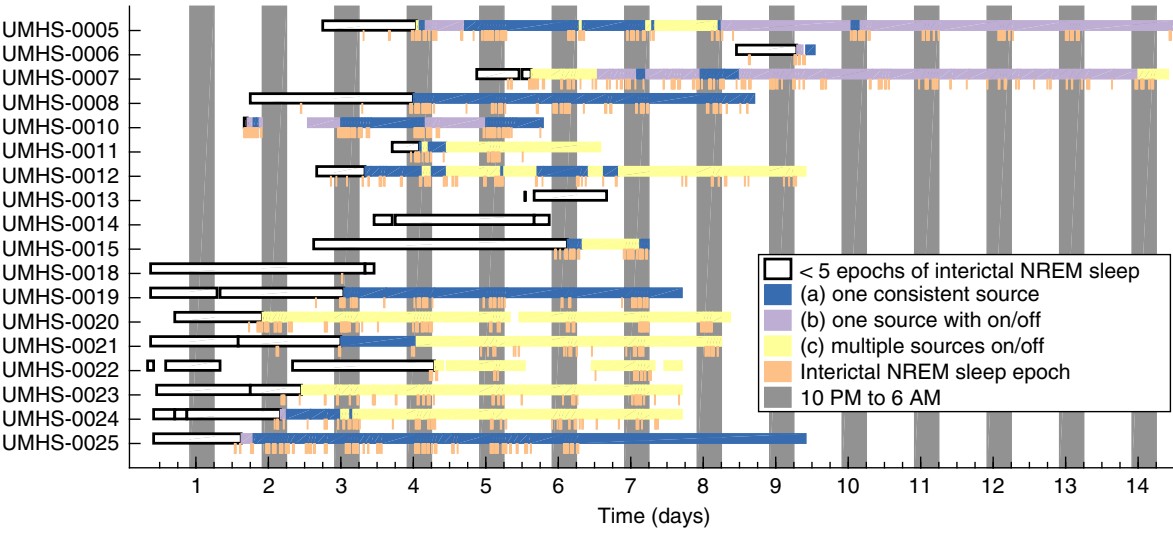

**Fig. 7** Category of variability as a function of total recorded time. Categorizations are made using all NREM interictal epochs from the recording start up to each given hour. Additionally, the location of the NREM interictal epochs are shown. Some patients cease to change categories as new data are acquired after a few days, whereas other patients still change category even after many days

patients. Additionally, comparison between the clinical data and the temporal variability of HFOs indicates that there are patients in whom HFOs should not be interpreted in isolation. Other confounds include the vast number of medication combinations and withdrawal schedules, sleep deprivation, unrecorded seizure types, undersampling with the electrode placement, and perioperative disturbances, all of which are likely to affect HFOs, as well as seizures. This unavoidable confound affects all patients, and argues strongly against choosing a single, short time window in which to evaluate HFOs. It is our primary conclusion that HFOs should be interpreted in context with all other available information regarding epileptogenic networks, each diagnostic tool providing additional pieces to the puzzle.

Identifying HFO rates as a "moving target" does not mean HFOs are not a biomarker of epilepsy. Our first hypothesis actually reinforces that HFOs are indeed strongly correlated with the SOZ. In fact, some of the variability in HFOs was actually due to variability in the epileptic networks. Two of the patients with multiple seizure foci illustrate this point. Patient UM-06 had bilateral hippocampal depth electrodes, with independent seizure foci on each hemisphere (and thus no resection), but HFOs only on the side with more frequent seizures. In patient UM-07, there were multiple groups of high-HFO channels that correlated with different daily seizure types in the temporal lobe, yet there was another focus that did not have high HFOs in the parietal lobe. Both regions (parietal lobe and temporal lobe) were resected and the patient has been seizure free. One might argue that it impossible to know whether both regions needed to be resected; however, it was clinically indisputable that the parietal region—which had a radiographic lesion but no HFOs—was an independent, clinically relevant focus, while the temporal lobe was also independently epileptogenic based on non-HFO markers. In both of these cases, HFOs would have been helpful but not solely sufficient to make clinical decisions.

Many researchers hope to use HFOs to reduce hospital stays by identifying seizure networks in a very brief period. Our results suggest that this is not straightforward, and in the patients studied here would not have been successful. Specifically, HFO rates in a 10-min segment of data cannot be assumed to be representative of HFO rates over the full recording, and even two days of recording is likely not enough to determine a stable answer. However, our data do suggest that HFOs can provide important

information regarding the underlying epileptic tissue even before seizures occur. For instance, the asymmetries were not found to be different when using short data segments versus all analyzed data per subject (i.e., 2 h per Mayo cohort, all interictal NREM sleep for UM cohort). This suggests that information from HFOs in short time segments is not wrong, but may be incomplete in some patients when viewed in isolation. We also observed interesting relationships between HFO rates and the ictal patterns (Fig. 5), suggesting that HFO detections are closely linked with underlying ictogenic activity, although the exact timing was not investigated in this study. It is possible that the HFO clusters move not because they are an unreliable biomarker of ictal activity, but because they are tightly coupled to underlying, unknown ictal generation and propagation networks, which also vary over time. The prime example is multifocal epilepsy, which has multiple seizure types arising from more than one region of the brain: such patients require special care to assure enough time has been allotted to capture the full extent of the seizure network (s). Even in patients with a single epileptic focus, each individual seizure may arise and/or involve from a sub-section of that focus. Thus, there appears to be a higher level of spatial and temporal resolution to ictal networks than the broad, binary categorization of tissue being epileptogenic or not. Unfortunately, accessing these underlying propagation networks is beyond current technological capabilities, though high resolution microwires[40] and microarrays[41] have found evidence of such effects. This situation further illustrates the need to incorporate as much clinical data—and recording time—as necessary when making surgical decisions.

It is also important to point out this work evaluates the use of NREM, interictal HFOs for patients undergoing extraoperative EEG to localize the seizure focus. We did not assess intraoperative HFO mapping, which is currently being used in a trial to tailor the extent of surgical resection of a focus that has already been identified[42,43].

The inclusion of the normal control group provides important data regarding the use of HFOs in epilepsy. While it has been previously reported that HFOs (both ripples and fast ripples) are present in this control group[37], this fact remains underrecognized. The current study now also demonstrates that there is considerable temporal variability in HFO location in control patients. We suggest that any process to utilize HFOs clinically

must account for this fact—there will always be a channel with "the highest HFO rate," even if the recording does not include the seizure onset zone. In addition, restricting the analysis to include only fast ripples is not sufficient to assure accuracy, as we detected fast ripples on 11/12 control patients' recordings. It is very possible that human supervision could have identified that there was not a clear seizure focus in these patients, but there is a strong potential for misdiagnosis if HFO rates are used in fully automated fashion or without experienced clinical correlation. Thus, although our first hypothesis concurs with past results that HFOs are highly correlated with SOZ, it is imperative to include more clinical information than simply identifying the location with the highest HFO rate, even if restricted to fast ripples.

Our data demonstrate that HFO patterns change even two weeks after implantation. This suggests that there is no universal time period in which HFOs would stabilize to a completely reliable solution. This is consistent with earlier work related to how many seizures are needed to determine the SOZ in the hospital[32], as well as recent work with long-term invasive monitoring, which showed that seizure patterns change even several weeks after implantation[44]. If HFOs are indeed biomarkers of seizures—which are themselves changing over time—it should not be surprising that HFOs patterns would also change. It would not be feasible to choose a single, controlled time period during monitoring that would be consistent for all patients. Therefore, no recommendation can be made at present regarding the amount of time needed for identifying epileptogenic tissue based on HFOs, just as recommendation cannot be made about the amount of time needed to observe the full distribution of a patient's seizure onset patterns. Both seizures and HFOs appear to be caused by non-stationary processes, and thus it is very difficult to determine whether the brain might behave differently outside of the recording window. However, as the median amount of recording time until the last change in categorization was about two days, (Fig. 7), it appears at least 2 days are required to obtain a relatively complete picture. At the very least, this longer period can help identify patients in which HFOs appear to be variable in both location and time. In patients with multiple seizure types/foci, further data are necessary to determine whether HFOs can enhance standard clinical decision-making.

These results utilized automated predictors rather than human review, which comprised the majority of past clinical HFO research. However, it is important to point out that our qHFO detection algorithm was previously validated with high concordance to human scoring and clinical outcomes[26,45]. In addition, the primary finding—that there are patients from all three categories—was verified by manual review, and the findings were consistent using two other automated detection schemes. Several of the patients were then verified to have multiple seizure foci based upon the clinical EEG reading. Thus, high confidence can be placed that the results represent the actual variability of the HFOs, not variability secondary to the detection process.

HFOs remain a promising biomarker of epilepsy, with potential to provide information about brain activity (both pathological and normal) and seizure-networks. Our results corroborate with the findings of many other publications which used smaller data sets: while HFOs are associated with epileptogenic tissue, HFOs are not a perfect biomarker—HFOs occur outside of epileptogenic regions and even in patients without epilepsy. Our results show that HFOs are even more associated with epilepsy than previously considered—in addition to their known spatial utility, in some patients they also display temporal changes linked with temporal changes in the seizure network. These results do raise the question of exactly what we expect HFOs to provide and how to best integrate their information with existing clinical procedures. The presence of HFOs outside the SOZ can represent either 'normal'

HFOs or areas that should be considered as part of the epileptic network. As there is currently no perfect method to determine whether an HFO is epileptic[7], clinical decisions should not depend solely upon the presence of HFOs, but should also include all other clinical data. Our results suggest that HFO rates should not be considered a simple "always-on" signal of epileptogenic tissue, just as traditional EEG signals are not. Simply put, in the perioperative period that characterizes a standard hospital admission for intracranial monitoring, "finding the channels with the highest HFO rate" is not a stationary biomarker. The complex temporal dynamics of HFO rates must be considered when interpreting HFO data, and short time-scale results may not be representative nor provide enough information to characterize the epileptogenic zone. Just as seizures spread beyond their onset region and sometimes change in location, HFOs may also move to other regions within or outside the propagation network under certain conditions. Identifying and understanding HFO locations will help characterize the full seizure network. Deciding where to resect is an extremely complex process, and every piece of information is needed. Instead of using HFOs as an alternative to that process, we suggest that HFOs can serve as an important adjunct to current methods, providing unique, interictal information about the epileptic network and potentially increasing the clinician's confidence in final decisions.

## Methods

**Patient population.** EEG data from patients who underwent intracranial EEG monitoring were selected from the Mayo Clinic and from the University of Michigan. Subjects selected from the Mayo clinic were those who met the following criteria as of June 2015: at least one night with consecutive, interictal data from 1 to 3 AM, with EEG sampling rate at least 5000 Hz. This yielded 103 patients, 91 of whom had epilepsy and 12 of whom had chronic facial pain. Subjects selected from the University of Michigan Hospital were those who met the following criteria as of October 2017: data from both intracranial EEG electrodes (for HFOs) and extracranial EEG electrodes (for sleep scoring), sleep scoring for at least one night completed by a certified sleep technician, and EEG sampling rate of at least 5000 Hz. One additional patient was excluded who had a hematoma between the grid and cortical tissue less than 24 h after implantation, disrupting the recording. This yielded 18 patients (16 adults and two children). The more stringent requirements applied to the Michigan patients could not be applied to the Mayo patients, as scalp (extracranial) EEG data were not available to allow sleep scoring. The 121 patients are divided into three cohorts: the Mayo Cohort includes 91 patients with epilepsy with interictal data recorded from one night, exactly 1–3 AM; the Control cohort includes 12 patients without epilepsy from Mayo Clinic, also with one night of data from exactly 1–3 AM; and the UM Cohort includes 18 patients recorded at the University of Michigan Health System, who had longer recordings and sleep scores available. Metadata regarding the seizure onset zone (SOZ) and surgery outcome were available for both the UM and Mayo cohort. The hours of 1–3 AM were selected as a time with high likelihood of non-rapid-eye-movement (NREM) sleep, since actual scoring of sleep was unavailable for the Mayo and Control cohorts. SOZ was determined by reading the official clinical report after the full hospitalization, and RV was determined in direct consultation with the neurosurgeons, who identified which channel locations were resected after comparing post-op MRI with pre-op clinical labels.

Mayo epilepsy and Control cohort data were recorded using a Neuralynx acquisition system (Bozeman, MT) with sampling rate of 32 kHz and a 9 kHz anti-aliasing filter[9]. Data for the UM cohort were recorded either using a Blackrock acquisition system (Salt Lake City, UT) with sampling rate of 30 kHz and a 10 kHz anti-aliasing filter[26,45], or a Natus Quantum acquisition system with sampling rate of 4096 Hz and ~1200 Hz anti-aliasing filter. All data from the Neuralynx and Blackrock systems were downsampled to 5 kHz in Matlab (Mathworks, Natick, MA) using the decimate function, which imposed a 2 kHz anti-aliasing filter.

All patients were adults or children with refractory epilepsy or chronic pain, and all patients underwent long-term intracranial monitoring. For the epilepsy patients, the monitoring was in preparation for resective surgery. All data were acquired with approval of local IRB and all patients consented/assented to share their deidentified data. Further details about the patient population are provided in Table 1, section A. For the UM patient cohort, the resected volume was determined based on official clinical reports, written by the treating neurologists and neurosurgeons, as well as individual review of each case with the treating neurosurgeons. For all epileptic patients, the SOZ was based on official clinical reports completed after the entire recording session. Patient ILAE outcome data are listed in Table 1, section B; note that the Mayo cohort had significantly more patients with poor outcomes, so it is difficult to compare results between the two centers. Interictal data were defined as more than 30 min from the start or end of

any seizure, based upon previous work showing HFOs change within that time period[39]. Note, when identifying segments of interictal NREM sleep, short microarousals (<1 min) were considered part of the NREM bout but HFOs during the arousal were redacted, whereas arousals > 1 min signaled a separation between NREM sleep segments. For this analysis, NREM sleep was defined as sleep stages 2–4. Data were processed using Matlab (Mathworks) and the GDFP program[26]. The total number of HFOs identified is shown in Table 1, section C. Note that the majority of the data are from UM, in which sleep scoring was performed.

**Experimental design.** Three specific hypotheses were tested: 1) HFOs are preferentially increased within the SOZ (2 h per Mayo cohort, all interictal NREM sleep for UM cohort), 2) the association of HFOs and SOZ is similar, on average, whether using 10-min data segments or all available data per patient, and 3) that the channels with the highest HFO rates are not consistent across varying 10-min segments. We utilized three cohorts of patients: all consecutive patients with epilepsy from Mayo Clinic meeting criteria; all consecutive patients with epilepsy from the University of Michigan (UM) meeting criteria; and all patients from Mayo Clinic with chronic facial pain (i.e., without epilepsy) meeting criteria. The facial pain group ($n = 12$) serves as the Control group without epilepsy, which is usually unavailable in studies of intracranial EEG. All data were analyzed in a pseudo-prospective fashion, in which the same parameters were applied to all patients.

**Computation of HFO rates.** HFOs were computed using three methods, and the full analysis was completed for each method. The first method is the qHFO method[26]. This algorithm uses a common average reference, applies a sensitive HFO detector (the "Staba detector")[35] and artifact detectors, and redacts HFO events coincident with detected artifacts. The Staba detector uses an 80–500 Hz bandpass filter, and identifies HFOs as oscillations with at least 6 peaks and with the smoothed, rectified signal being greater than five standard deviations above the background[35]. The artifact detectors specifically identify fast transients and non-focal events[26]. The artifact rejection method was previously found to have 79% agreement with human reviewers, which is comparable to interrater variability[26]. Previous work with the Staba detector was found to be indistinguishable from a human reviewer[37]. The second method (qHFO-FR) restricts the HFO detections to those within the "fast ripple" band, and is identical with the exception of using a 200–500 Hz bandpass filter instead of the 80–500 Hz range in the standard qHFO detector. The third method is quite distinct, and uses a Hilbert transform[24]. This last method was used to assure that the findings were not specific to our automatic detector, and is included on an open-source HFO-detection software package[46]. Of note, this last method is not well suited to analyzing long-term files under clinical conditions, as it does not remove artifacts and runs 20 times slower than real time on our system (our qHFO detector runs faster than real time on the same system). Software are available from the authors upon request.

For each HFO detector, the HFO rate (number of events per unit time per channel) was computed for each 10-min segment of data. Due to microarousals during NREM sleep, the amount of analyzed time per 10-min segment was, in some cases, less than 10 min. For visualization, the channels were grouped according to the SOZ and/or resected volume (when known) and then ordered such that the channels with similar HFO rates were nearby (using the MATLAB "optimalleaforder" function). Note that this order has no relationship with the original channel numbers from a given electrode configuration; it merely aids visualization of the different groups with similar HFO rate.

**Quantifying the relationship between HFO-rate and SOZ.** The value of HFOs as an epilepsy biomarker is based upon the fact that HFO rates tend to be higher within the epileptogenic zone. An objective way to measure this is to compare the average HFO rate in one region versus another with a retrospective tool. We previously used this "HFO rate asymmetry" to document how HFO rates are increased within the clinically determined SOZ[26]. The asymmetry is computed by first taking the average HFO rate over channels within and without the SOZ. The difference of these two quantities is then divided by the sum to form the asymmetry. The asymmetry values can be interpreted similar to correlation values: +1 implies all HFOs are in the SOZ (perfect "correlation"), −1 implies all HFOs are outside the SOZ (perfect "anti-correlation") and 0 implies that the mean rate is the same within and without the SOZ (no "correlation"). The asymmetries are computed either on all analyzed data per patient (2 h per Mayo cohort, all interictal NREM sleep for UM cohort), or on each 10-min segment of that analyzed data. Right-tailed Wilcoxon Sign-Rank tests were used to assess whether the median asymmetries were positive, whereas Wilcoxon Rank-Sum tests were used to assess differences in the median asymmetry between the distributions for the full and 10-min cases.

For a more localized analysis, in each 10-min segment we determined the single electrode that had the highest number of HFOs, for each of the three detectors. For the entire recording, we then reported the percentage of time each electrode was the "maximum channel," as well as how often any maximum channel was within the SOZ or resected volume. A perfectly stable result would be a single channel with 100% (Supplementary Fig. 5A,D,G), or that the maximum channel was within the SOZ (Supplementary Fig. 5B,E,H) or resected volume (Supplementary Fig. 5C, F,I) 100% of the time.

**Categorization of temporal variability.** As demonstrated in Fig. 1, we observed four categories of temporal variability: a) a single set of channels with consistently high HFO rates, b) a single set of channels with high HFO rates, but where the HFO rates are not high over all times considered; c) multiple sets of channels with high HFO rates, each of which has independent temporal evolution; and d) too few HFOs are recorded to determine the temporal variability. For sake of quantification, we define the "consistent" requirement for case (a) as at least 40% of the 10-min time epochs having an HFO rate greater than the mean rate over all time epochs, and that no more than 5% of the time epochs considered have zero HFOs detected. Likewise, we define "too few" for case (d) as all channels have less than 0.5 HFOs per minute over all times considered.

Next, we sought to develop an automated method to classify which of the four variability categories pertains to each patient. While case (d) can quickly be recognized and labeled, the other cases require several steps. These steps are described in detail below. First, the method determined how many independent regions (groupings of channels) exist (case (c) versus cases (a) and (b)), as well as the HFO rate per time epoch for each group. Second, the method applied the definition of category (a), in case only one group was found, to distinguish categories (a) and (b).

The process of determining how many independent groups of channels exist with similar temporal dynamics in HFO rates, while simultaneously determining the temporal dynamics of each group, is in the class of problems known as blind source separation. One of the main methods for blind source separation is non-negative matrix factorization (NMF), where the matrix (number of channels by number of 10-min epochs) of HFO rates ($R$) is factorized into a matrix describing the relative weight of each channel within each group of channels ($W$) and the effective HFO rate of each channel group over time ($H$), with $R = W \times H$[47]. Note, this construction assures that both the membership weights and effective HFO rates are non-negative. We selected units for $W$ and $H$ such that the mean effective HFO rate per time epoch (using matrix $H$) is set to equal the mean HFO rate of $W \times H$, i.e., the approximate overall HFO rate per epoch.

The NMF procedure alone cannot determine the number of channel groups, but rather finds the optimal channel groups and their temporal evolution for a specified number (denoted $K$) of groups. To determine the number $K$, we start with $K$ equal to the minimum of the following: the number of time epochs, the number of channels, and 12 (a reasonable upper bound). After applying the NMF algorithm with a given value of $K$, our method considers whether the Spearman correlation in time (rows of $H$) or space (columns of $W$) between signals is found to be greater than 30%. If so, the procedure is repeated with $K$ being set to $K$–1. In cases where the last value of $K$ is greater than one, the data are labeled category (c) (i.e., more than one cluster of channels is present). Otherwise, $K = 1$, resulting in either category (a) or (b). We then apply the consistency rule (definition of category (a)) to the one row of $H$. As the NMF is stochastic in nature, the entire categorization algorithm is repeated 10 times for all data, with the final category being the most frequently assigned category over the 10 repetitions. In case of a tie, the more complex category is assigned, with order of complexity being (c) most complex, (b), and (a) least complex. The algorithm was developed utilizing all interictal, NREM data for the first 10 subjects in the UM cohort, and then applied to all subjects.

**Manual validation.** The accuracy of the detector itself was previously validated to be indistinguishable from human reviewers[26,33]. Thus, the validation in the current work was to determine whether the overall results regarding the temporal variability were true. We verified the variability results manually in 8 subjects to ensure that the automated categorization reflects actual HFO location. One patient from each category (a)–(c) was selected at random from the UM cohort and from the Mayo cohort, plus two additional patients (categories (a) and (c)) from UM, for a total of 8 patients. Three 10-min epochs and the smaller of either 30 channels or all intracranial channels were selected. Epochs and channels were chosen based on the qHFO rates to give a representative picture of the automatically defined category. A board certified epileptologist experienced in HFO detection (WS) then manually observed the location of HFOs in each file using RIPPLELAB[46], software specifically designed for manual HFO scoring, and verified that the category based on the qHFOs reflected the HFOs viewed manually.

**Assessment of sufficient recording time.** Given the observed temporal variability, we considered how much recording time would be sufficient to observe a complete view of the channel groups and dynamics of the HFO rates. As a simple measure, we report if the categorization changes between different nights of 1–3 AM data in the UM cohort. As a more advanced measure, we simulated a possible clinical setting where at some given hour, the decision is made to stop the recording and interpret all the data recorded thus far. Utilizing the 10 UM cohort subjects, we simulated ending at each possible hour of the clock, and applied the categorization algorithm to all interictal, NREM data from the start of the recording until that time. We then report the resultant category for that time. We limited the analysis to NREM sleep to avoid the confounding factor of the state of vigilance; during periods without any new data the most recent determination was maintained.

**Statistical analysis**. This article focuses on reporting observational statistics, such as HFO rates and the fraction of patients in a given category. The number of patients in a given category is assumed to follow Poisson statistics, meaning that the variance is assumed equal to the number of counts. Binary comparisons between the fraction of patients in two given categories is completed using the $\chi^2$ statistic with one degree of freedom. Right-tailed Wilcoxon Sign-Rank tests were used to assess whether median asymmetries were positive, and Wilcoxon Rank-Sum tests were used to compare distributions of HFO-rate asymmetries.

**Data availability**. The HFO data and algorithms have been deposited in the Deep Blue Data repository at the University of Michigan with the identifier https://doi.org/10.7302/Z29K48F3[48].

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

## Acknowledgements

We gratefully acknowledge all those involved with obtaining the data, including all those at the Mayo Clinic and the University of Michigan. Research reported in this publication was supported by the National Institute of Health (NIH) Big Data to Knowledge Initiative under grant number K01-ES026839 (S.G.), the NIH National Institute of Neurological Disorders and Stroke (NINDS) under grant numbers K08-NS069783, R01-NS094399, Doris Duke Foundation Clinical Scientist Development Award #2015096 (W.S.), and UH2-NS095495, R01-NS063039, R01-NS092882 (to G.W., B.B.).

## Author contributions

S.G., Z.I., C.C., G.H., B.B., O.S., H.G., G.W. and W.S. performed data collection and reviewed the manuscript. S.G. performed data analysis. S.G. and W.S. prepared the manuscript.

## Additional information

**Competing interests:** The authors declare no competing interests.

