## [Peer Review File · Nature Communications]

Reviewers' comments:

Reviewer #1 (Remarks to the Author):

This paper studies the consistency of HFO rates over periods of hours to days in 101 epilepsy subjects and 12 patients without epilepsy (implanted with intracranial electrodes for electrical stimulation treatment of chronic facial pain). HFOs are potential biomarkers for determining the epileptogenic zone, which can be surgically removed to treat patients that are medically refractory. There are three unique approaches in this study: (i) first study to use long durations of recordings compared to prior art; (ii) first study to include a control group; and (iii) first study to include a large group of patients relative to prior art. The authors state that the goal of the present work is to “use HFO data from a large cohort of patients with and without epilepsy to determine the stability over time of the HFO rate within a single sleep stage, in preparation for large-scale clinical translation.”

To conduct their study, they use the qHFO algorithm to detect HFOs (which removes some artifacts), analyzed NONREM sleep (sleep stages annotated by experts) when available, and data from 1-3AM without regards to sleep stage. Comparisons between NONREM specific and 1-3AM were done with UM data and not Mayo data since no sleep stage annotations were available for Mayo data.

I appreciate the magnitude of this study and believe that a rigorous study of HFOs is needed right now given how much “hype” HFOs have received in the epilepsy community. However, I am not sure what the final message of this paper is and I think it needs some clarification.

HFOs do move around but move around in same way that seizures move around..

Specific Comments:

- The results using HFOs basically suggest that sometimes seizure onset zone (SOZ) correlates to electrodes exhibiting highest HFO rates, and sometimes not. Is there any interesting feature that all the patients share where there is good agreement with high HFO rates and SOZ? For example, are they all temporal lobe patients? Are they all lesional patients etc?
- For broad reader-what is difference between epileptogenic zone and SOZ in main paper? I think you define SOZ as RV?
- What is the distribution of patient type perhaps defined by location of electrode implantation (temporal lobe, frontal lobe etc)? It would be useful to understand this distribution as the different types of patients may exhibit different HFO trends (say according to your four categories a-d).
- Should clinicians look at HFOs? Did the clinicians look at the highest HFO channels for failed patients to see if they missed something? What if you predict surgical outcome based on agreement between highest HFO regions and clinically annotated SOZ – that is, if the agreement is high, you predict a successful outcome, and if the agreement is low you predict a failed outcome. A good algorithm would have high agreement for success cases and low rate for failed cases. The latter could prove helpful to clinicians, to alert them of potential failures.
- Medications vary over days? How do you account for different medical states to account for variability in HFOs?
- Fig. 1: UM-10 there is a region not resected that had larger HFO rates..was this patient a success? I may have missed this but would be good to clearly state the outcomes of the example patients you include in your figures.
- To a broad reader who is not well-versed in epilepsy, may need to explain why you are looking during NONREM sleep only? What about inter-ictal wake data? I understand there are more artifacts during wake and REM sleep...is this why you study sleep?
- Was RV determined with post-operative MRIs or the neurosurgeon drawing lines on a brain image? How precisely is RV defined.

- You select top channels as those with max peak. How about top 5% of channels. Would a less stringent definition of top channel change conclusions?
- Were all detected HFOs compared to clinically annotated HFOs or were not all data annotated?
- When studying the spatiotemporal variability in HFO rates in a large cohort of patients, authors applied blind source separation algorithm to group electrode channels by temporal dynamics of HFOs (non-negative matrix factorization). They then match groups to spatial locations of the electrodes and find that HFO groups are spatially close to one another. Why is this surprising when these channels are more likely to be functionally connected?
- Can you add to Fig7 and show results for patients with other Engel scores. Maybe also split by patient type/lobe (see comment above)? What I am getting at here is that it would be a more complete study if you can explain the variability in HFO in relation to SOZ. Maybe HFOs work well for some patient types and not others. Else my personal take away is that this is a provocative study that shows that HFOs are actually not that helpful.
- The authors point out on line 281 that the HFOs were not present in lesioned area. Perhaps it would be useful to describe more scenarios (if any) where HFO analyses may be insufficient.
- The authors state on line 309 that the HFOs should not be interpreted in isolation but rather in conjunction with all other information. I guess I am not convinced that HFOs are helpful at all in determining the SOZ or EZ. What I would be very interested in is whether HFOs can “flag” a clinician as to what they may have missed? Since HFO results have not been summarized for failure patients, this is hard to say if it is true.
- Authors state on lines 336-337 that HFO rates can exhibit interesting relationships to ictal patterns even hours before seizure events, but it does not appear to be in majority of the patients studied. Again, which patient types is typically true for?
- When quantifying whether highest HFO rate channels correspond to SOZ, a percentage is used. But shouldn't you take into account how many electrodes are in the RV with respect to how many are implanted? For example, isn't a 100% agreement in a patient with 100 electrodes implanted and 4 electrodes in RV a much more precise agreement than 100% agreement in a patient with 10 electrodes implanted and 4 electrodes in RV a much more precise agreement? Perhaps consider a degree of agreement statistic that takes into account all electrodes, RV and RV complement etc.
- One concern is that all these results depend on automatically detected HFOs not corroborated by hand annotations. I understand it will take forever to annotate such big data, but the results are just not so convincing with the exception that the clearest message appears to be that HFO rates are not reliable in general even if recorded over long periods. Further, the much appreciated control group also shows similar characteristics as the epileptic group with regards to HFO variability. This is a comment that you don't have to address necessarily but what are your thoughts?
- Artifacts: what artifacts does the qHFO detect and what artifacts does it miss? Does it deal with muscle artifacts, background fluctuations?

Reviewer #2 (Remarks to the Author):

In this manuscript, the authors apply an automated HFO detector to a large population (> 100) of patients. They provide the first assessment of spatiotemporal variability in HFO rates in a large population of human subjects. They show that in most patients HFOs exhibit variability in both space and time. This result calls into question the usefulness of HFO for surgical planning. The manuscript is well-written, and the detailed procedure is well-described. However I do have two main comments, which would hopefully help improve the impact of the manuscript.

Main comments.

1. My main concern is simply stated: what if the HFO detector is not working? If we assume that the detected HFOs are valid, then we conclude that HFOs are highly variable in both space in time. Because we trust the HFO detector, we believe this variability reflects changes in each patient's "true HFO" which we're trying to observe in the noisy brain data. Although the authors are careful not to state this directly, in this case, I interpret their results to mean that HFO is not a useful biomarker; it's too noisy and unreliable for surgical planning. This conclusion challenges the large existing literature that HFO is a useful biomarker of SOZ or RV. Therefore, to support it, requires an overwhelming amount of compelling evidence. I don't think that the current manuscript reaches this level of evidence.

Here are some ideas to bolster the authors conclusions:

a. The authors choose a specific automated HFO detector. But there are many different options for automated HFO detectors, here are three,

1) Ellenrieder, von, N., Andrade-Valença, L. P., Dubeau, F., & Gotman, J. (2012). Automatic detection of fast oscillations (40-200 Hz) in scalp EEG recordings. *Clinical Neurophysiology* : 123(4), 670–680.

2) Zelmann, R., Mari, F., Jacobs, J., Zijlmans, M., Dubeau, F., & Gotman, J. (2012). A comparison between detectors of high frequency oscillations. *Clinical Neurophysiology* , 123(1), 106–116. <http://doi.org/10.1016/j.clinph.2011.06.006>

3) Blanco, J. A., Stead, M., Krieger, A., Viventi, J., Marsh, W. R., Lee, K. H., et al. (2010). Unsupervised classification of high-frequency oscillations in human neocortical epilepsy and control patients. *Journal of Neurophysiology*, 104(5), 2900–2912. <http://doi.org/10.1152/jn.01082.2009>

If the authors choose a subset of these existing measures – the most popular, or those in wide use, that have been validated, and not necessarily the examples above – and repeat the analysis, do they find similar results? If so, that would add confidence that the variability in the HFO is not due to the choice of HFO detection method.

b. The authors assume that the variability in the detected HFOs reflects variability in the "true HFOs" that occur in a patient's brain. What if the variability is due to the detection method? Or, how do we know that the variability is due to the changes in the patient's HFO? Is there any sense of measure variability for the HFO detection method? Perhaps there is a simulation or surrogate analysis that could be done to explore the variability we would expect from this measure when analyzing noisy brain voltage data.

c. As it stands, the HFO detections appear too variable to be clinically useful. Is there a way to examine a meaningful subset of HFOs, and show that those selected HFOs provide useful clinical evidence? What if only HFO at seizure onset, or near seizure onset, or at some other meaningful time were examined? Would that help improve the reliability? I think this is important to show that the HFO detection method used "works" in a best-case scenario.

d. In line 124-125, the method is forced to pick a max HFO channel, even if that max is not significantly higher than the other electrodes. Could that be a problem? I would guess that this choice introduces noise by selecting a random channel when there is no HFO in the interval. It seems like this choice would impact sensitivity.

2. The authors do not state that HFO is a poor biomarker of SOZ. But, that seems to be the main result. Are there positive results that add confidence to this negative result?

a. The authors provide one example patient where the variability may have meaning (Line 200-202). That is an interesting conclusion, but this connection appears to only have been made for one subject. Could the conclusion that the variability has meaning be further developed? Do the patients in

category (c) with multiple sources tend to have worse surgical outcomes?

b. Lines 326-331. "Looking at all the results, one possible interpretation is that the HFO channel groups are not providing information just about the epileptogenic zone, but also a subset of the pathological ictal and interictal activity propagation network. Thus, just as seizures spread beyond their onset region, the HFO channel groups might also extend beyond the epileptogenic zone under certain conditions." That's a nice idea. Is it possible to show that the variability in the HFO is meaningful in these data and for these patients?

c. Lines 337-338. "For instance, we observed interesting relationships between HFO rates and the ictal patterns (Fig. 5)," This relationship is just for one patient. Can that interpretation be further supported by additional analysis of other patients?

d. Could the localization percentage be analyzed to show that it's closer to 0 for Engel class 2 or higher? That might show that the HFO method is "working" in some sense.

Other comments:

Line 25,26. That's not good. In "several" out of 113 patients? What about the other patients with variable HFO but one foci?

Line 27. "These results reaffirm the strong correlation between HFOs and tissue initiating seizures," How "reaffirm"? These results seem to call into question the strong correlation between HFOs and SOZ?

Line 101. It does not seem to be accurate to say "NREM sleep" in all subjects. Only one cohort of subjects had sleep staging.

Line 122-125. I interpret these results to show HFO is not a useful biomarker. Most of the time, the channel with the max HFO is not in the SOZ or RV. That seems to contradict previous work.

Line 149-151. Please provide more details. Does "completely different channels" mean no overlap at all? How many manual verifications per patient? What was the maximum number, what was the minimum number?

Line 246-247. "We observe that HFO variability is not limited to the resected volume or SOZ, but often extends across a wider region." That seems like a big problem. It seems to show either (1) HFO is not a useful biomarker of SOZ or RV, or (2) the HFO detector used here isn't working, or there's some problem with the method. How can we distinguish between the two?

Line 296-297. "Many prior clinical reports have shown strong correlations of HFO rate to the seizure onset zone". How is that possible, given the variability shown here? Did those other authors get lucky?

Line 301. "We find that HFOs are correlated with SOZ as reported previously," Where is this positive result shown in this paper?

Line 303-305. "We thus observe that HFOs are not simply an "always on" signal for identifying epileptic tissue but exhibit complex temporal dynamics that appear variably related to behavioral state," Isn't the state fixed at 1-3 AM sleep?

Line 363-365. "HFOs therefore provide insight into epileptic networks and activity, but their spatial

distribution, temporal dynamics and their relationship to focal seizure generation and epilepsy is likely quite complex." Or, HFOs don't provide insight? The simplest interpretation of the results reported here seems to be that HFO is not a useful biomarker of SOZ or RV.

Line 386-387. "It is well known that some patients demonstrate different ictal patterns over the course of admission; our data suggest that HFO patterns also change concordantly," This is not shown in the population that HFO patterns change with different ictal patterns. It's only shown in one example.

Methods. How many patients have Engel class 1?

Line 490. Only 2 patients were selected for manual review. Because the results are so controversial, and challenge existing understanding in the field, I wonder if the gold-standard of manual inspection will be necessary for more patients.

Line 535-537. How was the visual inspection of the grouping performed? What is "a large number of Mayo cohort and Control subjects" ?

Figure 5. It's difficult for me to make sense of this figure. The seizure type seems to evolve from Type 2 to Type 3 to Type 5. But it's not clear how the HFO rates evolve.

Reviewer #3 (Remarks to the Author):

In this manuscript, the authors present observational findings of lack of spatial and temporal consistency of HFO occurrences. The data set used is impressive, with a large number of subjects (113) and a set of 12 control patients with focal pain. This is an important topic relevant for clinical translation of HFOs. However, the study is limited by being primarily observational, with no clearly articulated hypothesis or statistical outcomes, although it does make the point that interictal HFOs are highly variable to the point of being potentially unreliable for clinical use.

Major critiques:

1. The use of an automated detector is reasonable, especially given the quantity of data processed. However, please include statistics on the accuracy of the method, as it is critical to the paper - references to prior literature are helpful but not sufficient. Missing information includes types of filters used and artifact detection methods. It would be helpful if the software were made available to readers either as supplementary information or as an online toolbox.
2. A major conclusion as stated in the abstract and text, that HFOs are consistently linked to seizure generating sites, is not supported by the data as presented. The variability in localization percentages is striking, and some data sets show no HFOs in the SOZ (lines 238-245). Further, the maximal channel is outside the seizure onset zone (SOZ) the majority of the time.
3. The HFO variability may be impacted by cases with multiple seizure types and onset locations. One example is given, but in order to understand the implications of HFO variability in location percentage and over time, this information needs to be made available for the entire data set, and the relationship of HFOs to the different onset sites needs to be quantified.
4. Except for a single patient example, the interictal/ictal status of the chosen data segments is not

described. The methodology suggests that this was not taken into account, as timing was selected entirely on the basis of sleep status, while in some cases apparently continuous testing was performed (judging by figures). Seizures would introduce potentially high variability into the HFOs detected that could have impacted the results. Also see next point.

5. Limited information provided regarding the number of data segments analyzed per patient and how many segments were analyzed with sleep staging (all the UM data sets or just the two patients described?). The assumption that 1-3 AM is NREM sleep, which was applied to a majority of the recordings, is not likely valid, as intracranially implanted patients often have disrupted sleep. It is in fact possible to estimate sleep stage from intracranial data, with several recent publications describing this procedure, and this should either be done, or the discussion of limitation to NREM sleep should be dropped as it may not be relevant - especially considering that in a clinical setting, sleep staging is unlikely to be conducted as a prerequisite to HFO detection. I realize the authors were trying to minimize the confounding effects of state changes, but there are many other important potential confounds that are not considered, e.g. anticonvulsant medication changes/abortive medication dosing, and whether the analyzed data segments contained seizures or were immediately before or after a seizure.

6. Some patients were studied across multiple two hour time segments, whereas others were studied for a single two-hour segment (implied in lines 206-207). How does this factor into conclusions about temporal variability?

7. Data interpretation is highly descriptive, with little or no quantification and over-reliance on anecdotes. For example, it is not clear how terms such as "temporal variability", "high HFO rate", and "reaching stability" are measured or defined. Even in the anecdotes presented, details are lacking, e.g. how different seizure onsets differ in lobe/sublobe of origin or electrographic onset pattern, or whether they were organized into clusters as is commonly the case, which could explain the temporal patterns described. This is likely related to the fact that there is no hypothesis presented that the data analysis is designed to test, nor an outcome measure.

8. Line 218 states no difference between epilepsy patients and the controls was found. No difference in what - HFO rates? Temporal variability? If HFO rates do not differ between patient and control groups, would that not undermine the entire thesis that HFOs are biomarkers of seizure-initiating brain sites? For that matter, the data appear to be an indictment of the concept of using interictal HFOs to supplant seizure recordings in invasive EEG cases. The results of this analysis show striking temporal and spatial variability in HFO occurrence, and this seems to contradict prior studies demonstrating strong concordance of HFOs detected in 10 minute time segments with the SOZ, and as predictors of surgical outcome. This is stated directly in the third paragraph of the discussion, but further elucidation/discussion of what is likely to be a highly controversial conclusion is needed.

9. The question of how many days are required to to obtain stable HFO information (which I assume means no additional sites detected) is similar to studies examining the number of seizures required for intracranial monitoring to detect all seizure onset sites. See for example Struck et al *Epilepsia* 2015 as well as older literature (Blume and Sadler 1994). This link should be acknowledged, and techniques used for these prior studies may be useful here. However - how was this determined in this paper from the limited two-hour time segments analyzed, especially as we don't know how many patients had multiple time segments to analyze? It appears this analysis was only done on two UMN cases.

10. The statement in the discussion that "ictal generation and propagation networks" vary over time is interesting, but entirely unsupported. Please discuss the physiological and/or clinical basis for this statement, and cite relevant literature.

11. The investigators took care to confirm that the variability finding is present regardless of frequency band, and after results are confirmed by human review (in a limited sampling from two cases). However - what were the results of the visual review, and how did the variability compare quantitatively to the automated detection results?

12. Not clear that HFO rate is the best measure to assess clinical relevance of the HFO biomarker. Some interictal discharges occur more rarely than twice per minute, and yet may provide critically important clinical information, while normal phenomena may be accompanied by HFOs and occur more frequently. The paper then, can only conclude that use of HFO rates picked up by an agnostic detector alone may be unreliable - not that HFOs themselves are.

13. Qualitative statements that need to be supported with data & statistical testing:

Line 180: "quite close together"

Line 245: "qualitatively similar."

Line 251: "often extends across a wider region"

Lines 257-258: "very common for the answer to change from one night to the next"

Lines 268-9: "some patients...sufficient data were recorded....continued to have changes".

Line 286: "Other patients also had significant changes in HFO rate..."

Line 499: "agree with"

These are just some examples. The use of unsupported qualitative statements occurred throughout the text, detracting considerably from readability.

Minor:

1. The title and abstract should specify whether interictal or interictal + ictal HFOs are under consideration.
2. The reason that the prior studies selected NREM sleep is likely that interictal discharges are more frequent during this time, and because movement artifacts are minimized. As this topic is discussed in methods, this should be mentioned also.
3. Please define abbreviations at first use (e.g. Mayo, UM, SOZ).
4. Please provide references and justification for the frequency definition of ripples vs. fast ripples.
5. The investigators speculate that the intracranial implantation itself causes HFO rates to vary over time. This is an intriguing hypothesis, but it is not testable with this dataset nor are prior literature cited to support it. It should therefore be presented as a speculation for future study, rather than established fact.

General response to all reviewers:

We thank all the reviewers for their constructive comments and interest in this work. Below, we respond to each concern individually, but wish to comment to the group on two over-riding concerns from all reviewers.

First was the desire for better validation. In response to this general concern, we have made substantial additions to this work. In particular, 1) we have added 8 additional patients from UMich, all of whom have multiday recordings (adding > 1300 hours of continuous EEG), 2) we have re-analyzed the entire dataset of (now) 121 patients with a completely different HFO algorithm, and 3) we have manually validated the most important finding of the work, that there are patients in whom the location of peak HFOs changes over time, in a total of 8 patients.

Another general concern was what should be done with these results, i.e. does this mean HFOs are invalid? To this we reply emphatically “No!” HFOs are clearly a marker of epileptic tissue, as has been shown by many others and as we now reaffirm in the paper. We agree that our results argue strongly against picking 10 minutes of interictal data and basing a resection upon those HFO rates. Clinicians are rightfully skeptical of this idea, as it seems to discount decades of clinical experience. But there is clearly a strong biomarker signal, which we have tried to show more clearly in this new version—HFO data have tremendous potential! The issue is how to use them. We believe there is a much more reasonable clinical option, which strangely seems never to have been suggested before—HFOs should be interpreted alongside all the other clinical data (i.e. EEG spikes/seizures, MRI, SPECT, PET, etc). We have made many changes to the paper to make this point stronger.

As a comparison, epileptologists have been dealing with multimodal data for years, and sometimes it is nonconcordant. But we do not reject MRI, PET and SPECT studies if they show data outside our hypothesis, nor do we reject spikes and seizures if they change over the course of the hospitalization. They are all pieces of the puzzle. It is wrong to assume that just one modality, whether it be HFOs or anything else, can stand alone as a biomarker. But it would also be wrong to ignore any of them—they all have their role if used correctly. This is how we believe HFOs should be used. We hope this paper adds to the evidence of the role and proper usage of HFOs.

We have made significant changes to the abstract, methods, and conclusions to state first that, like the past work on HFOs, our data show that HFOs are clearly correlated with the SOZ. We show that selecting arbitrary 10-minute epochs gives similar qualitative answers to using the entire dataset, and that both are significantly correlated with SOZ. We then discuss how HFOs should be incorporated with other clinical modalities—i.e. including all clinical information to make the decision, not just a brief segment of 10 minutes of data. As we now state in the conclusion, “HFOs remain a promising biomarker of epilepsy...Our results show that HFOs are even more associated with epilepsy than previously considered...HFO rates should not be considered a simple “always-on” signal of epileptogenic tissue, just as traditional

EEG signals are not... Deciding where to resect is an extremely complex process, and every piece of information is needed. Instead of using HFOs as an alternative to that process, we suggest that HFOs can serve as an important adjunct to current methods, providing unique, interictal information about the epileptic network and potentially increasing the clinician's confidence in final decisions."

Reviewer 1:

1. I appreciate the magnitude of this study and believe that a rigorous study of HFOs is needed right now... However, I am not sure what the final message of this paper is and I think it needs some clarification. HFOs do move around but move around in same way that seizures move around.

This comment is echoed by the other reviewers. We have rewritten large portions of the paper and included specific hypotheses to make our overall conclusion more clear: that HFOs are indeed biomarkers of epilepsy, but they are not necessarily stationary in time, just like all of the other EEG biomarkers (spikes, slowing, seizures, etc). Effectively, HFOs are part of the answer but not the whole answer--and certainly not in isolation of all other clinical data. We believe that the proper role of HFOs is to "serve as an important adjunct to current methods, providing unique, interictal information about the epileptic network and potentially increasing the clinician's confidence in final decisions" [conclusion]. Thus, we now state in the abstract that "robust HFO interpretation requires prolonged analysis in context with other clinical data, rather than isolated review of short data segments."

2. The results using HFOs basically suggest that sometimes seizure onset zone (SOZ) correlates to electrodes exhibiting highest HFO rates, and sometimes not. Is there any interesting feature that all the patients share where there is good agreement with high HFO rates and SOZ? For example, are they all temporal lobe patients? Are they all lesional patients etc?

We found no correlation between clinical metadata (pathology, location, gender, age, etc) and the classification. Our data now includes five patients that ended up having multifocal epilepsy, and we show that many, but not all, of them have multifocal HFO sources. The paper now states that "In the UM NREM cohort, where more metadata were available, we also compared the categorizations with age, gender, pathology, SOZ location, and duration of recording. No statistically significant associations were found using any of the HFO detectors ($p > 0.05$, Kruskal-Wallis and Fisher's Exact Test)." We also report the numbers of multifocal patients in each category. It is important to point out that having the HFO data at the time of clinical diagnosis would have been informative in each case, but only in the setting of each patient's clinical context. Please see the answer to #13 for more information.

Note that our new data proves the HFOs are highly correlated with SOZ. The issue is whether the specific channels are stable...and whether EVERY channel is within the SOZ. More details on this follow below.

3. For broad reader-what is difference between epileptogenic zone and SOZ in main paper? I think you define SOZ as RV?

The reviewer points out a somewhat confusing distinction, and we acknowledge that it is important to avoid jargon for the broad reader: EZ is the theoretical region that has to be resected for seizure freedom, SOZ (also sometimes called the Ictal Onset Zone) is what the clinicians think/hope is the EZ based upon the monitoring, RV is what actually gets resected. These three regions are rarely the same, but admittedly that gets confusing. We have added this explanation to the Introduction: “...the region of brain responsible for generating seizures, known as the epileptogenic zone. However, there is no perfect method to identify the true epileptogenic zone. Standard clinical practice is to place intracranial electrodes and observe spontaneous seizures, with the patient remaining in the hospital for many days. Clinicians estimate the true epileptogenic zone by determining the Seizure Onset Zone (SOZ), which guides surgical resection. The final Resected Volume (RV) is not necessarily the same region, as it is limited by anatomical and functional considerations.” After this initial description, we are careful to use SOZ to describe what was identified clinically, and epileptogenic zone to describe the true focus that needs to be resected.

4. What is the distribution of patient type perhaps defined by location of electrode implantation (temporal lobe, frontal lobe etc)? It would be useful to understand this distribution as the different types of patients may exhibit different HFO trends (say according to your four categories a-d).

As we stated in #2, there was no correlation between lobe and categorization of HFO variability per patient, and with 121 patients it is not feasible to list all information in table 1. We believe that the issue is not that certain lobes are more/less likely for HFOs to “work”, but that patients can have multifocal networks, just like spikes and seizures are often unstable over time during an EMU stay. It is thus not surprising that HFOs—as valid biomarkers of the active seizure network—would also move around.

5. Should clinicians look at HFOs? Did the clinicians look at the highest HFO channels for failed patients to see if they missed something? What if you predict surgical outcome based on agreement between highest HFO regions and clinically annotated SOZ – that is, if the agreement is high, you predict a successful outcome, and if the agreement is low you predict a failed outcome. A good algorithm would have high agreement for success cases and low rate for failed cases. The latter could prove helpful to clinicians, to alert them of potential failures.

We wholeheartedly agree! We think it’s time to start showing HFOs to clinicians as part of their analysis, rather than treating them like a black box completely devoid of clinical information. We believe this paper will be a springboard to start doing work such as this reviewer suggested. We did not perform any such trials in this work, however. Note that we did compare outcomes with HFO rates in our earlier paper describing our HFO detection and artifact reduction algorithms (Gliske, et al., 2016). The current paper now concludes with the statement that

“HFO rates and their temporal variability over many days can serve as an important adjunct to current methods, providing unique, interictal information about the epileptic network and potentially increasing the clinician's confidence in final decisions.”

6. Medications vary over days? How do you account for different medical states to account for variability in HFOs?

This is a good point that we did not analyze. Given the nearly infinite combinations of med doses/ combinations/ withdrawals in addition to other seizure induction maneuvers such as sleep deprivation, it would not be feasible to analyze this robustly with this dataset. However, it is specifically this complicated situation that motivates the need for not looking at simply 10 minutes of data—we already know there are many temporal variables in the EEG—why should we think HFOs are immune to this confound? We have added this point to the Discussion “Another confound is the vast number of medication combinations and withdrawal schedules, sleep deprivation, and perioperative disturbances, all of which are likely to affect HFOs as well as seizures. This unavoidable confound affects all patients, and argues strongly against choosing a single, short time window in which to evaluate HFOs.... It would not be feasible to choose a single, controlled time period during monitoring that would be consistent for all patients.”

7. Fig. 1: UM-10 there is a region not resected that had larger HFO rates..was this patient a success? I may have missed this but would be good to clearly state the outcomes of the example patients you include in your figures.

We have fixed the labels to make this clear. Mayo 040 had no resection, UM-07 and UM-10 both had Class I outcomes.

8. To a broad reader who is not well-versed in epilepsy, may need to explain why you are looking during NONREM sleep only? What about inter-ictal wake data? I understand there are more artifacts during wake and REM sleep...is this why you study sleep?

We have clarified that the primary reason is that “Slow wave sleep was chosen because it has fewer muscle artifacts and the rate of HFOs is higher.” And we chose to follow this pattern to allow “us to compare directly with previous work”. This is important because otherwise these results could be discounted as analyzing different data.

9. Was RV determined with post-operative MRIs or the neurosurgeon drawing lines on a brain image? How precisely is RV defined.

The methods now state “SOZ was determined by reading the official clinical report after the full hospitalization, and RV was determined in direct consultation with the neurosurgeons, who identified which channel locations were resected after comparing post-op MRI with pre-op clinical labels.” Note also that we have changed the figures, and now none of the results are directly dependent upon RV.

10. You select top channels as those with max peak. How about top 5% of channels. Would a less stringent definition of top channel change conclusions?

Adding even more channels actually makes things worse, as we showed in our 2016 paper—if you force the algorithm to pick channels, it will always do so, even when it makes no clinical sense. That leads to even more false positives. However, we have removed the analysis of Max channel in response to other concerns. We felt the previous Fig. 2 was too confusing to the overall message. Our current paper focuses on the asymmetry in the HFO rate, which is essentially the percentage of total mean HFO rate that occurs within the seizure onset zone. We use this measurement to prove that the HFOs are indeed good biomarkers of epilepsy, but in this work we do not try to define seizure onset zone based upon HFOs.

11. Were all detected HFOs compared to clinically annotated HFOs or were not all data annotated?

This analysis detected > 1 million HFOs with just our qHFO algorithm, not to mention the other two algorithms. It is impossible to validate all of them. We now reinforce in the intro that “The algorithm was previously validated to yield similar results as a human reviewer, and allows us to analyze HFO features in vast datasets while avoiding interrater variability.” However as described above and throughout this Response, we have gone to considerable effort to validate the findings. In particular, in 8 patients an expert reviewer manually scored HFOs to determine which category each patient belonged to. We found that each patient was matched with the category determined by our categorization in Fig. 6.

12. When studying the spatiotemporal variability in HFO rates in a large cohort of patients, authors applied blind source separation algorithm to group electrode channels by temporal dynamics of HFOs (non-negative matrix factorization). They then match groups to spatial locations of the electrodes and find that HFO groups are spatially close to one another. Why is this surprising when these channels are more likely to be functionally connected?

We apologize for this misunderstanding. We actually did not analyze the spatial proximity of the HFOs. The reviewer is perhaps referring to our explanation of the visualization tool we used, which groups HFO channels based upon their firing rates to make the figure easier to analyze. We now clarify that “Note that this order has no relationship with the original channel numbers from a given electrode configuration; it merely aids visualization of the different groups with similar HFO rate.” One of the reasons for including the sequence of Fig.1 (HFO rate over time), Fig 3 (blind source separation), then Fig 4 (visualization of the electrodes on a brain schematic) with the same four patients was to show what the actual data looked like.

That being said, the reviewer brings up a good point of functional connectivity and spatial organization of HFOs. This is an under-explored topic that we hope to evaluate in future work with this same dataset.

13. Can you add to Fig7 and show results for patients with other Engel scores. Maybe also split by patient type/lobe (see comment above)? What I am getting at here is that it would be a more complete study if you can explain the variability in HFO in relation to SOZ. Maybe HFOs work well for some patient types and not others. Else my personal take away is that this is a provocative study that shows that HFOs are actually not that helpful.... The authors point out on line 281 that the HFOs were not present in lesioned area. Perhaps it would be useful to describe more scenarios (if any) where HFO analyses may be insufficient... The authors state on line 309 that the HFOs should not be interpreted in isolation but rather in conjunction with all other information. I guess I am not convinced that HFOs are helpful at all in determining the SOZ or EZ. What I would be very interested in is whether HFOs can “flag” a clinician as to what they may have missed? Since HFO results have not been summarized for failure patients, this is hard to say if it is true... Authors state on lines 336-337 that HFO rates can exhibit interesting relationships to ictal patterns even hours before seizure events, but it does not appear to be in majority of the patients studied. Again, which patient types is typically true for?

We have removed the previous Fig. 7 based upon other comments, and now focus our overall results on a more straightforward conclusion: that HFOs are useful but should be considered another biomarker similar to spikes and evaluated over longer periods. We have added 8 more UM patients to the analysis to enrich our long-term results (new fig. 7), and now analyze the relationship between categorization, ILAE score, and lobe of the brain involved. We found no correlation between metadata and patient category. We also could not find any way to identify when patients will be like UM-07 (switching after prolonged periods) based solely upon HFOs. However that scenario is precisely the point we are trying to make. The clinical suspicion on UM-07 was that there was another seizure type not yet captured...which is precisely what prompted his recording continuing for several days longer than normal. In that case, the clinical scenario was absolutely essential in judging the HFOs. Trying to base clinical decisions devoid of such information would be misguided. Our goal with this work was to show that 10-minutes of HFO data is not enough to give the “whole answer.” We now state throughout the paper that these results suggest HFOs should be used alongside other clinical data as part of the decision process. Future work will explore how this can be optimized.

However, we appreciate the concern shown by this and the other reviewers regarding what to do with these results. We would like to expand somewhat to show this reviewer some examples. These are vignettes showing a range of extreme outcomes from our UM cohort. These are not enough to publish but show the varying effects the reviewer asked for. Note that if one just reads the three bylines (in a,b,c below) it may seem disappointing, but the “rest of the story” in each case shows how applying HFOs to clinical practice can work. (Note that HFO data were not available at the time of any of these clinical decisions—we have just outlined how HFOs could have been used in each case.)

- a) Pt 7, a patient with two foci, only one with HFOs, both resected, class I outcome.
 - The HFO-silent parietal focus had an MRI lesion, and none of the temporal seizures (which were associated with HFOs) were his typical seizure. He

finally had a typical parietal seizure, affirming that the lesion had to come out. There was much discussion about whether the temporal lobe also needed to be resected. Seeing the rampant HFOs there would have reinforced the need to resect.

- b) A patient (UMHS 0018) with two independent foci (Left temporal, Left occipital) both with HFOs, only L temporal resected, class I outcome.
 - The unresected occipital lesion was felt to be asymptomatic and too close to eloquent cortex. In this case, it was crucial to see the patient have a seizure from the occipital lobe to see that it was not clinically concerning. There is no way HFOs alone could have determined this. However, the presence of HFOs left behind leaves us worried she may not remain seizure free.
- c) A patient with two foci both with HFOs, one resected, class III outcome.
 - The second focus was unresectable, but was felt to be heavily involved based upon clinical evaluation. HFOs would have reinforced that the surgery was unlikely to be wholly successful.

Thus we think there is still much utility in HFOs when used in clinical context: they can warn about surgeries unlikely to be successful (C) and can reinforce when the resection should be expanded (A). Conversely, applying clinical information can show us when HFOs can be safely ignored (B). In all, it is clear that HFOs can be highly valuable if evaluated in clinical context. We believe that one of the major barriers to translation of HFOs is that clinicians distrust the idea that 10 minutes of interictal HFO data could supplant a week of recording. We also distrust it!! With this paper we show that such distrust is well placed...but that does not mean HFOs have no value. For this paper, we focus on re-establishing the utility of the biomarker (Fig. 2) while drawing attention to the need for longer recordings. We hope this will lead the field to incorporating HFOs into all other clinical information.

As we conclude in the paper: “Deciding where to resect is an extremely complex process, and every piece of information is needed. Instead of using HFOs as an alternative to that process, we suggest that HFOs can serve as an important adjunct to current methods, providing unique, interictal information about the epileptic network and potentially increasing the clinician's confidence in final decisions.”

14. When quantifying whether highest HFO rate channels correspond to SOZ, a percentage is used. But shouldn't you take into account how many electrodes are in the RV with respect to how many are implanted? For example, isn't a 100% agreement in a patient with 100 electrodes implanted and 4 electrodes in RV a much more precise agreement than 100% agreement in a patient with 10 electrodes implanted and 4 electrodes in RV a much more precise agreement? Perhaps consider a degree of agreement statistic that takes into account all electrodes, RV and RV complement etc.

We agree with this and several other concerns about that confusing figure. It is very difficult to come up with a fair metric that can show the utility of HFOs across so many patients. We have

changed this analysis to utilize just the asymmetry value, which compares the average rate of of HFOs within versus without the SOZ.

15. One concern is that all these results depend on automatically detected HFOs not corroborated by hand annotations. I understand it will take forever to annotate such big data, but the results are just not so convincing with the exception that the clearest message appears to be that HFO rates are not reliable in general even if recorded over long periods. Further, the much appreciated control group also shows similar characteristics as the epileptic group with regards to HFO variability. This is a comment that you don't have to address necessarily but what are your thoughts?

We have added additional hand verification, now in 8 patients, to verify the 3 classes. We find additional evidence (Fig. 2) that HFOs are related to SOZ, but in total our results show that decisions should not be based solely on HFO rates. In the case of control patients, they can be considered the example of what might happen if clinical context is lost: e.g., even in a patient with epilepsy it is possible the electrodes are in completely the wrong place—there is ALWAYS a peak HFO channel. It is just too risky to determine the SOZ from 10 minutes of interictal data. Thus it is important to see HFOs over time, and in the future we will also need to establish ways to determine “what is abnormal and what is not.” For the control cases, the rates of HFOs tend to be quite low, though sometimes epileptic patients have similar values. The first key is to find electrodes with anomalously-high rates, as we detailed in our original algorithm paper (2016 Clin Neurophys). That method would have shown, for instance, that the control patient did not have a group of electrodes clearly different from the rest. The next key will be to identify features that are specific to epilepsy. Fast ripples are somewhat helpful, but are not nearly as specific as hoped. Our future work is to find better features.

16. Artifacts: what artifacts does the qHFO detect and what artifacts does it miss? Does it deal with muscle artifacts, background fluctuations?

Our artifact detector is described in the 2016 paper. The goal was to remove obvious transients (pops) that produce false HFOs, and remove diffuse oscillations that occur across all channels, which are likely due to line noise and are not localizing. More subtle artifacts are still too difficult to remove accurately. However, we found that our algorithm has 85% agreement with human reviewers in identifying artifacts. The methods now state that “The artifact detectors specifically identify fast transients and non-focal events”, and we make several statements to describe the artifact reduction method and point back to the validation in that previous paper.

One major problem with artifacts is that some are not obviously artifact versus unusual brain activity. Our algorithm leaves such events as HFO detections. Thus the main artifact that is still present is when muscle is picked up by the intracranial leads. This produces a low amplitude “oscillation” that can be picked up as an HFO. We are currently working on a solution for this, but it is challenging unless the surface EEG is available. This is much less common during slow wave sleep, which is another reason our data were limited to that time period. Background fluctuations are accounted for by the Staba HFO algorithm, which compare

each HFO with the current background.

Reviewer #2

17. My main concern is simply stated: what if the HFO detector is not working? If we assume that the detected HFOs are valid, then we conclude that HFOs are highly variable in both space in time. Because we trust the HFO detector, we believe this variability reflects changes in each patient's "true HFO" which we're trying to observe in the noisy brain data. Although the authors are careful not to state this directly, in this case, I interpret their results to mean that HFO is not a useful biomarker; it's too noisy and unreliable for surgical planning. This conclusion challenges the large existing literature that HFO is a useful biomarker of SOZ or RV. Therefore, to support it, requires an overwhelming amount of compelling evidence. I don't think that the current manuscript reaches this level of evidence.

The reviewer brings up two points: validation of the detector and the conclusion of how to use HFOs in clinical practice based upon our results. We have made significant additions to the paper to answer these concerns, detailed below.

18. The authors choose a specific automated HFO detector. But there are many different options for automated HFO detectors, here are three,

- 1) Ellenrieder, et al (2012). Automatic detection of fast oscillations (40-200 Hz) in scalp EEG recordings. Clinical Neurophysiology : 123(4), 670–680.*
- 2) Zemann, R., et al. (2012). A comparison between detectors of high frequency oscillations. Clinical Neurophysiology , 123(1), 106–116. <http://doi.org/10.1016/j.clinph.2011.06.006>*
- 3) Blanco, J. A., et al (2010). Unsupervised classification of high-frequency oscillations in human neocortical epilepsy and control patients. Journal of Neurophysiology, 104(5), 2900–2912. <http://doi.org/10.1152/jn.01082.2009>*

If the authors choose a subset of these existing measures – the most popular, or those in wide use, that have been validated, and not necessarily the examples above – and repeat the analysis, do they find similar results? If so, that would add confidence that the variability in the HFO is not due to the choice of HFO detection method.

We have redone the entire analysis using another detector. The three examples chosen were actually not well suited for this analysis: 1 is for scalp EEG, 2 is extremely slow and has very high false positive rate in uncurated data such as this, and 3 is actually the literal predecessor of our own detector, though it requires considerable parameter fitting for each patient that is hard to reproduce and not amenable to large scale use (which is why we developed the qHFO algorithm). We chose the following algorithm, which uses a Hilbert transform (Crepon B, Navarro V, Hasboun D, Clemenceau S, Martinerie J, Baulac M, Adam C, Le Van Quyen M. Mapping interictal oscillations greater than 200 Hz recorded with intracranial macroelectrodes in human epilepsy. Brain. 2010;133(Pt 1):33-45.) We chose this one because it is so well known

that it is one of the included algorithms in RIPPLELAB (Navarrete et al. PLoS One, 2016), and will be readily available to readers. We found that this detector finds many HFOs that are likely due to noise (not as well correlated with SOZ). However, it correctly identifies the SOZ in the UM patients, and also comes to the same conclusions: there are many patients with moving SOZ. These results, which validate those of our detector, are now part of Fig. 2 and 6. More importantly, the real question here is whether there are patients in which the HFO answer changes over time. We also now have human validation of 8 patients, corroborating each of the three classes. These results validate our detector's results: there are indeed patients with single consistent sources, single sources that flicker on and off, and multiple HFO sources.

As a side note, the comparison algorithm takes > 20 times as long as our algorithm, and that analysis alone took 3 CPU years to process the dataset used in this manuscript—this is the primary reason it took so long to resubmit this paper. It would not be suitable for this analysis on a large scale.

19. The authors assume that the variability in the detected HFOs reflects variability in the “true HFOs” that occur in a patient’s brain. What if the variability is due to the detection method? Or, how do we know that the variability is due to the changes in the patient’s HFO? Is there any sense of measure variability for the HFO detection method? Perhaps there is a simulation or surrogate analysis that could be done to explore the variability we would expect from this measure when analyzing noisy brain voltage data.

As stated above, we have added a human validation on 8 patients, verifying that in each case the patient’s HFOs “did what the algorithm said they did.” We also have corroborating results from a completely different automated detector. Thus we are confident that the result of having multiple sources is not due to noise, though it is certainly possible that noise (or any subtle effects on signal quality) would affect the sensitivity of detections over time and might produce on/off phenomena. That potential noise effect is actually one of the reasons longer recordings are necessary. Additionally, the fact that the HFOs are highly correlated with SOZ (Fig. 2) further supports that what we detect is actually highly related to epileptic tissue.

20. As it stands, the HFO detections appear too variable to be clinically useful. Is there a way to examine a meaningful subset of HFOs, and show that those selected HFOs provide useful clinical evidence? What if only HFO at seizure onset, or near seizure onset, or at some other meaningful time were examined? Would that help improve the reliability? I think this is important to show that the HFO detection method used “works” in a best-case scenario.

We have not evaluated whether specific HFO timing might be more valuable; this is a very good idea for future work. However, such ideas are themselves clear departures from the current standard of 10 minutes of data. The fact that the reviewer is suggesting these ideas corroborates our conclusion that 10 minutes is not enough to fully characterize the HFO signal. However, with our new results (Fig. 2) reinforcing that the HFOs are, indeed, strongly correlated with the SOZ, we are trying to make it clear that HFOs are not failed biomarkers—they clearly have diagnostic value and we believe the best way to figure out how to use them is to

recapitulate the process by which EEGs were originally evaluated: showing them to clinicians in their clinical environment, rather than as isolate biomarkers in brief data snippets.

In effect, we believe HFOs are similar to epileptic spikes, though perhaps more specific to the SOZ based upon several papers by Gotman and others. Spikes are crucial in clinical decisions, but no clinician would ever base a decision upon 10 minutes of interictal spike data. Some early work tried to show that interictal spikes were highly localizing, but this was later disproven. Even seeing a single seizure is rarely trusted for full characterization of the SOZ. HFOs are likely a similar situation. Thus, we strongly disagree with the idea that our results invalidate HFOs. On the contrary, what is often the case is that patients actually truly have variability in their SOZ and the HFOs are correctly identifying this. We have added information and discussion regarding 5 patients with multiple foci to illustrate this point.

21. In line 124-125, the method is forced to pick a max HFO channel, even if that max is not significantly higher than the other electrodes. Could that be a problem? I would guess that this choice introduces noise by selecting a random channel when there is no HFO in the interval. It seems like this choice would impact sensitivity.

We agree and that was one of our desired discussion points. However, that part has been removed so that we could focus on more straightforward hypotheses with the asymmetry measurement.

22. The authors do not state that HFO is a poor biomarker of SOZ. But, that seems to be the main result. Are there positive results that add confidence to this negative result?

Due to the several concerns about whether our results “disproved HFOs as a valid biomarker”, we altered the paper considerably to first focus on the fact that HFOs are indeed highly correlated with the epileptic tissue.

We have added a new analysis to address this point, the new Fig. 2. The goal of this figure is to prove our first hypothesis, that HFOs are actually valid biomarkers of the SOZ, and that similar results are obtained whether 10-minute segments or the entire recording is used. Our analysis used the asymmetry value, which compares the ratio of HFOs inside the SOZ versus outside the SOZ. A perfect biomarker would have asymmetry = 1, and a random biomarker with no correlation would be 0. Our results show that the HFOs are preferentially within the SOZ. The UM data were more significant, likely because they were limited to true NREM sleep. Our results also show that a random 10 minute epoch is likely to give similar results to utilizing data from the whole hospitalization. The problem with the 10-minute segments is that they are often just a portion of the answer, and their precise distribution is variable. Thus HFOs are highly correlated with SOZ, but the precise answer that is given can change between 10-minute periods.

23. The authors provide one example patient where the variability may have meaning (Line 200-

202). *That is an interesting conclusion, but this connection appears to only have been made for one subject. Could the conclusion that the variability has meaning be further developed? Do the patients in category (c) with multiple sources tend to have worse surgical outcomes?... .*

“Looking at all the results, one possible interpretation is that the HFO channel groups are not providing information just about the epileptogenic zone, but also a subset of the pathological ictal and interictal activity propagation network. Thus, just as seizures spread beyond their onset region, the HFO channel groups might also extend beyond the epileptogenic zone under certain conditions.” That’s a nice idea. Is it possible to show that the variability in the HFO is meaningful in these data and for these patients? ... “For instance, we observed interesting relationships between HFO rates and the ictal patterns (Fig. 5),” This relationship is just for one patient. Can that interpretation be further supported by additional analysis of other patients?

We have added in additional patients from UM, which allowed us to analyze 5 patients with multiple seizure foci. We find a mixture of class a, b, and c, even in this small cohort. We have also looked at the ILAE scores and find no clear relationship with seizure class. In short, there does not appear to be a clear ‘rule’ by which one can use HFOs to avoid performing the traditional EEG analysis. We conclude that HFOs tend to behave like spikes and seizures—they can move around and must be evaluated in similar fashion in their full clinical context, using all other clinical data. For more specific anecdotes, please see our answer to #13. We have moved this thought to the conclusion “Just as seizures spread beyond their onset region and sometimes change in location, HFOs may also move to other regions within or outside the propagation network under certain conditions. Identifying and understanding HFO locations will help characterize the full seizure network.”

24. Could the localization percentage be analyzed to show that it’s closer to 0 for Engel class 2 or higher? That might show that the HFO method is “working” in some sense.

We find no relationship between the asymmetry index (vs. SOZ) and ILAE outcome. But we now show that across all patients, regardless of outcome, there is clear correlation. HFOs are indeed good biomarkers of the SOZ, but should not be used in isolation.

25. Line 25,26. That's not good. In "several" out of 113 patients? What about the other patients with variable HFO but one foci?

We now confirm in 8 patients, at least two from each category, that the algorithm correctly identified each of the three classes. We have improved our description of our validation process. We also have included analysis of patients with multiple foci, showing that they sometimes only have a single HFO region. In short, different patients are different, and need to be evaluated in their clinical context.

26. Line 27. “These results reaffirm the strong correlation between HFOs and tissue initiating seizures,” How “reaffirm”? These results seem to call into question the strong correlation between HFOs and SOZ?... Line 122-125. I interpret these results to show HFO is not a useful

biomarker. Most of the time, the channel with the max HFO is not in the SOZ or RV. That seems to contradict previous work.....

As stated in the intro to this Response, and in several other of the previous questions, we have changed Fig. 2 to deal with this question, and have altered the text extensively to answer this question. We have removed the section about max HFO to concentrate on those other points. (note that we did already show that max HFO was unreliable in our 2016 paper).

27. Line 101. It does not seem to be accurate to say "NREM sleep" in all subjects. Only one cohort of subjects had sleep staging.

Thank you for pointing that out, we have edited the paper to make it more clear that the Mayo cohort is not necessarily NREM. We have also added 8 more patients with sleep scoring to increase the power of that analysis.

28. Line 149-151. Please provide more details. Does "completely different channels" mean no overlap at all? How many manual verifications per patient? What was the maximum number, what was the minimum number?

This sentence is no longer in the paper after our changes. However, we have expanded the human validation section and added patients as described in #25, and we have changed the description of the different channel groups (i.e. Fig 3) to make this more clear.

29 Line 246-247. "We observe that HFO variability is not limited to the resected volume or SOZ, but often extends across a wider region." That seems like a big problem. It seems to show either (1) HFO is not a useful biomarker of SOZ or RV, or (2) the HFO detector used here isn't working, or there's some problem with the method. How can we distinguish between the two?

See #26. Note that the variability is in the 10 min data segments, and observing for longer periods gives a fuller picture. Variability over different 10-minute data segments does not necessarily imply that HFOs are not a useful biomarker. Our overall conclusion is to use as much data as possible and incorporate the other clinical modalities.

30. Line 296-297. "Many prior clinical reports have shown strong correlations of HFO rate to the seizure onset zone". How is that possible, given the variability shown here? Did those other authors get lucky?... Line 301. "We find that HFOs are correlated with SOZ as reported previously," Where is this positive result shown in this paper?

Figure 2 shows that 10 minutes of data give similar answers (i.e. HFOs are preferentially within the SOZ), just not the complete, stationary picture. We believe our results corroborate prior data, though in this larger and longer cohort we have been able to identify variability in the precise location, not merely a predilection to be within SOZ.

31. Line 303-305. "We thus observe that HFOs are not simply an "always on" signal for

identifying epileptic tissue but exhibit complex temporal dynamics that appear variably related to behavioral state," Isn't the state fixed at 1-3 AM sleep?

This sentence no longer appears in the paper. However, the 1-3 AM data are not necessarily sleep. More importantly, this particular sentence was misleading. We meant to say that the brain activity changes dynamically, in any state. Those changes can potentially affect HFOs. We now use similar language to make a different statement in the conclusion "Our results suggest that HFO rates should not be considered a simple "always-on" signal of epileptogenic tissue, just as traditional EEG signals are not."

32. Line 363-365. "HFOs therefore provide insight into epileptic networks and activity, but their spatial distribution, temporal dynamics and their relationship to focal seizure generation and epilepsy is likely quite complex." Or, HFOs don't provide insight? The simplest interpretation of the results reported here seems to be that HFO is not a useful biomarker of SOZ or RV.

See #26. It is critical to note that the majority of HFOs are within the SOZ, as we now show in Fig. 2, even if the precise location is variable and sometimes contains channels outside the SOZ intermittently.

33. Line 386-387. "It is well known that some patients demonstrate different ictal patterns over the course of admission; our data suggest that HFO patterns also change concordantly," This is not shown in the population that HFO patterns change with different ictal patterns. It's only shown in one example.

We have removed this statement.

We have also added the analysis of five patients that ended up having multifocal epilepsy, and we show that many, but not all, of them have multifocal HFO sources. It is important to point out that having the HFO data at the time of clinical diagnosis would have been informative in each case, but would have had to be interpreted in each patient's clinical context. Please see the answer to #13 for more information about multifocal patients.

34. Methods. How many patients have Engel class 1?

We have added data regarding the ILAE classification of each patient that had such data available in Table 2. It is impossible to validate, but one reason why the Mayo HFO data may not be as robust in Fig. 2 is that there were many more patients with poor outcomes, which means the resected tissue did not comprise the full epileptogenic zone.

35. Only 2 patients were selected for manual review. Because the results are so controversial, and challenge existing understanding in the field, I wonder if the gold-standard of manual inspection will be necessary for more patients.

We agree, and have increased the manual review to 8 patients. We show that in each case the manual review agreed with the algorithm (at least 2 each of class a, b, and c).

36. Line 535-537. How was the visual inspection of the grouping performed? What is "a large number of Mayo cohort and Control subjects" ?

We apologize for the confusion, that statement was referring to the initial training, and no results were based upon that visual inspection. Our validation is now with manual testing in 8 subjects. We have revised this text.

37. Figure 5. It's difficult for me to make sense of this figure. The seizure type seems to evolve from Type 2 to Type 3 to Type 5. But it's not clear how the HFO rates evolve.

We apologize for the confusion. The reviewer correctly notices the seizure type evolving. Our point is to show how the HFO clusters change at the same time. To do this, we include bars that are the same as the HFO clusters from Fig. 3D2, spread out to fit this time scale. In effect this figure shows the seizure types and HFO clusters spread over time. We have clarified this figure in the text and caption.

Reviewer #3:

38. the study is limited by being primarily observational, with no clearly articulated hypothesis or statistical outcomes, although it does make the point that interictal HFOs are highly variable to the point of being potentially unreliable for clinical use.

We have changed the format of the paper to address three hypotheses: "1) HFOs are preferentially increased within the SOZ (2 hours per Mayo cohort, all interictal NREM sleep for UM cohort), 2) the association of HFOs and SOZ is similar, on average, whether using 10-minute data segments or all available data per patient, and 3) that the channels with the highest HFO rates are not consistent across varying 10-minute segments.." (methods)

Major critiques:

39. The use of an automated detector is reasonable, especially given the quantity of data processed. However, please include statistics on the accuracy of the method, as it is critical to the paper - references to prior literature are helpful but not sufficient. Missing information includes types of filters used and artifact detection methods. It would be helpful if the software were made available to readers either as supplementary information or as an online toolbox.

We added the data for the detectors in the methods under "computation of HFO rates." We have also added data on the number of HFO detected in Table 3. We now state that the

software is available to readers upon request. The software was originally published in our 2016 paper, and we have shared it with many groups already.

40. A major conclusion as stated in the abstract and text, that HFOs are consistently linked to seizure generating sites, is not supported by the data as presented. The variability in localization percentages is striking, and some data sets show no HFOs in the SOZ (lines 238-245). Further, the maximal channel is outside the seizure onset zone (SOZ) the majority of the time.

We have changed the hypotheses and figures to show that the HFOs are indeed biomarkers of the SOZ (Fig. 2). We have removed the analysis of max channel.

41. The HFO variability may be impacted by cases with multiple seizure types and onset locations. One example is given, but in order to understand the implications of HFO variability in location percentage and over time, this information needs to be made available for the entire data set, and the relationship of HFOs to the different onset sites needs to be quantified.

This addresses one of the main motivations for this manuscript—that both HFO rate (and thus the variability) can be affected by many things, including multiple seizure foci and onset locations. While subject UM 07 provides an interesting case to use an example, our focus is to demonstrate that such variability exists and to suggest that HFOs should be interpreted in context with the full clinical context. Unfortunately, it is not possible to include the localization data for all patients in this fashion. Aside from the difficulty in characterizing and even displaying results from this many patients, this question really requires a gold standard of the true epileptogenic zone, which is not available. The only gold standard is to compare resected volume in patients with perfect outcomes, but the RV is always much bigger than the HFOs suggest. Answering the question fully really requires a clinical trial.

However, we agree that more rigor is needed in analysis of multifocal patients. To address this and other concerns, we have increased our UM cohort from 10 to 18 patients, which coincidentally (these are consecutive patients) increased the number of multifocal patients from 2 to 5. We include the results of those multifocal patients in Table 4. For a further discussion of these multifocal patients, see responses #2, #13. We also now include specific text stating that no association was found between onset location and HFO variability category. We find that there is no consistent “answer”. While it would be satisfying to know that the variability was found only in patients with multiple foci, that is simply not the case. Conversely, the patients with multiple foci do not always have variability. In essence our data show that “anything can happen.”

While this may seem discouraging, we believe this is due to irrational expectations towards HFOs. It is simply too much to ask one biomarker to capture, in just 10 minutes, what we already know cannot be reliably captured in a whole week of recording. But the data are still important, just like all other EEG data are important. So our goal with this paper is to show that HFOs should still be used, but just in a different way. In general our conclusion is “Just as seizures spread beyond their onset region and sometimes change in location, HFOs may also

move to other regions within or outside the propagation network under certain conditions. Identifying and understanding HFO locations will help characterize the full seizure network. Deciding where to resect is an extremely complex process, and every piece of information is needed. Instead of using HFOs as an alternative to that process, we suggest that HFOs can serve as an important adjunct to current methods, providing unique, interictal information about the epileptic network and potentially increasing the clinician's confidence in final decisions."

42. Except for a single patient example, the interictal/ictal status of the chosen data segments is not described. The methodology suggests that this was not taken into account, as timing was selected entirely on the basis of sleep status, while in some cases apparently continuous testing was performed (judging by figures). Seizures would introduce potentially high variability into the HFOs detected that could have impacted the results. Also see next point.

We apologize for this confusion. All data in this paper are from interictal data, which we define as "more than 30 minutes from the start or end of any seizures." We have made that more clear throughout the paper and added it to the abstract.

43. Limited information provided regarding the number of data segments analyzed per patient and how many segments were analyzed with sleep staging (all the UM data sets or just the two patients described?). The assumption that 1-3 AM is NREM sleep, which was applied to a majority of the recordings, is not likely valid, as intracranially implanted patients often have disrupted sleep. It is in fact possible to estimate sleep stage from intracranial data, with several recent publications describing this procedure, and this should either be done, or the discussion of limitation to NREM sleep should be dropped as it may not be relevant - especially considering that in a clinical setting, sleep staging is unlikely to be conducted as a prerequisite to HFO detection. I realize the authors were trying to minimize the confounding effects of state changes, but there are many other important potential confounds that are not considered, e.g. anticonvulsant medication changes/abortive medication dosing, and whether the analyzed data segments contained seizures or were immediately before or after a seizure.

We agree that the 1-3 AM data is not ideal, though it is all that is available from Mayo. Estimates of sleep stage from the intracranial data, given the vast array of types and locations of electrodes, will still be subject to similar uncertainty in a dataset this large. We also apologize for the confusion regarding sleep staging. All patients in the UM cohort had sleep staging by a certified sleep technician, and we analyze both the manually marked NREM sleep and the 1-3 AM data in the UM cohort. We have revisited the manuscript and clarified the language to better indicate NREM sleep results vs. 1-3 AM results. Additionally, we have added 8 additional patients from UM, where NREM sleep is verified. The results are actually more robust in the UM data. We computed the results from 1-3 AM from the 18 UM patients, as a surrogate of the Mayo data, and found no significant difference versus using NREM sleep in the same cohort. Thus we believe that despite the limitations of the Mayo data, these results are valid. Most importantly, as shown in our new Table 3, 75% of the total number of HFOs in this paper are from the UM data. We now point out these issues in the text of the paper.

44. Some patients were studied across multiple two hour time segments, whereas others were studied for a single two-hour segment (implied in lines 206-207). How does this factor into conclusions about temporal variability?

There was no correlation between the number of segments, nor any other patient metadata, with the results. Our conclusion is that HFOs must be viewed in clinical context of each patient. The results now state “In the UM NREM cohort, where more metadata were available, we also compared the categorizations with age, gender, pathology, SOZ location, and duration of recording. No statistically significant associations were found using any of the HFO detectors ($p > 0.05$, Kruskal-Wallis and Fisher’s Exact Test).”

45. Data interpretation is highly descriptive, with little or no quantification and over-reliance on anecdotes. For example, it is not clear how terms such as "temporal variability", "high HFO rate", and "reaching stability" are measured or defined. Even in the anecdotes presented, details are lacking, e.g. how different seizure onsets differ in lobe/sublobe of origin or electrographic onset pattern, or whether they were organized into clusters as is commonly the case, which could explain the temporal patterns described. This is likely related to the fact that there is no hypothesis presented that the data analysis is designed to test, nor an outcome measure.

In response to this concern, we have made significant changes throughout the paper to make it more hypothesis driven, mathematically precise, and quantitative.

- a. We have modified the paper to be more clear that the four examples shown are not anecdotes, but are comprehensive demonstrations of our data analysis, which simply cannot be displayed on a per-patient basis with a dataset this large. These examples are presented first as an explanation, then the data are analyzed specifically in the “Occurrence rate of each category of spatiotemporal variability” section.
- b. We have modified the description of “temporal variability” to be mathematically precise in the methods “Categorization of temporal variability” section. Similarly, “reaching stability” is defined in “assessing sufficient recording time to converge on a solution” section. The assessment of “high HFO rate” is no longer a key measurable since our analysis focuses solely on asymmetry between mean rates inside and outside of SOZ.
- c. We have added specific hypotheses to test. “1) HFOs are preferentially increased within the SOZ (2 hours per Mayo cohort, all interictal NREM sleep for UM cohort), 2) the association of HFOs and SOZ is similar, on average, whether using 10-minute data segments or all available data per patient, and 3) that the channels with the highest HFO rates are not consistent across varying 10-minute segments.” We specifically define how to measure those values (the statistical tests in Fig. 2 and 6).
- d. We have added information on localization and other metadata, and show it has no effect on the results

- e. We have clarified that the data are all interictal, so concerns about seizure clusters are not pertinent.

In all, the overall goal is to show that HFOs need to be evaluated in clinical context, i.e. not in purely methodical fashion.

46. Line 218 states no difference between epilepsy patients and the controls was found. No difference in what - HFO rates? Temporal variability? If HFO rates do not differ between patient and control groups, would that not undermine the entire thesis that HFOs are biomarkers of seizure-initiating brain sites? For that matter, the data appear to be an indictment of the concept of using interictal HFOs to supplant seizure recordings in invasive EEG cases. The results of this analysis show striking temporal and spatial variability in HFO occurrence, and this seems to contradict prior studies demonstrating strong concordance of HFOs detected in 10 minute time segments with the SOZ, and as predictors of surgical outcome. This is stated directly in the third paragraph of the discussion, but further elucidation/discussion of what is likely to be a highly controversial conclusion is needed.

We apologize for this misleading statement. That statement meant that individuals without epilepsy have HFO variability. Prior publications from multiple research groups have already established that individuals without epilepsy have HFOs, though the rates are usually lower. We have completely reworked that section. We now verify (in agreement with past work) that HFOs are biomarkers of epilepsy: we show that 10 minute segments of NREM sleep are very likely to have most HFOs within the SOZ. We did not assess prediction of surgical outcome in this paper. However, we do agree with the statement that HFOs should not supplant seizure recordings. Both epilepsy and normal patients have temporal variability, and without clinical context HFOs can be misinterpreted. We feel our revised discussion and additional data (more UM patients, analysis of multifocal patients, assessing the two hypotheses, etc) now make this more clear. This increase in clarity should also make the conclusion less controversial.

47. The question of how many days are required to obtain stable HFO information (which I assume means no additional sites detected) is similar to studies examining the number of seizures required for intracranial monitoring to detect all seizure onset sites. See for example Struck et al Epilepsia 2015 as well as older literature (Blume and Sadler 1994). This link should be acknowledged, and techniques used for these prior studies may be useful here. However - how was this determined in this paper from the limited two-hour time segments analyzed, especially as we don't know how many patients had multiple time segments to analyze? It appears this analysis was only done on two UMN cases.

As noted, this analysis cannot be done in the 1-3 AM data, and instead we use the UM NREM sleep data (10 patients in the last version of the manuscript). The amount of data analyzed per patient is visible in new Fig. 7 (old Fig. 8). We agree that we needed more data to make this conclusion, and added 8 additional UM patients (new Fig. 7) to show that HFOs need to be assessed over time, just like seizures are.

We have included discussion of Struck et al. and thank the reviewer for this suggestion. While the analysis of how much data may be needed for the HFO variability category to not change has some similarity to the mentioned papers, there are several specific differences. Our event is more complex, as we are considering whether acquiring more data would change the interpretation of the full data set, i.e., change the category of HFO variability. The papers mentioned address how much data (seizures) is needed to see the full range of onset sites. Rather than fitting statistical models of the times of our “events” as were done in those papers, we instead choose to present the raw data, i.e., the specific times the categorization changing as more data is acquired. This avoids assumptions about whether a single statistical model can accurately fit the data, given the many confounding factors. We feel this presentation also will be more easily understood by a broader audience. Lastly, the chosen presentation keeps the focus on our hypotheses (that variability does exist) and our main conclusion (that HFOs need to be analyzed in concert with the other clinical information).

We also now mention data from the Neurovista study (Cook et al 2013) showing that after implantation there is prolonged change in the EEG signal. Of note, that group’s upcoming work shows that it takes 100 days (!) to stabilize.

48. The statement in the discussion that "ictal generation and propagation networks" vary over time is interesting, but entirely unsupported. Please discuss the physiological and/or clinical basis for this statement, and cite relevant literature.

We agree that we do not have any supporting data for the statement. We have softened the statement to read that it is “possible”. The problem is that current EEG technology cannot see these effects, so we would have to speculate on the physiology. However, we now cite literature on two high-res technologies (microwires and Utah array) that found evidence for this type of effect (Stead et al 2010, Schevon et al 2012).

49. The investigators took care to confirm that the variability finding is present regardless of frequency band, and after results are confirmed by human review (in a limited sampling from two cases). However - what were the results of the visual review, and how did the variability compare quantitatively to the automated detection results?

In response to this concern for validation, we expanded the human review to include 8 patients, at least one from each class from each center, and provide more details in the text regarding how the human review was conducted. We find that manual review agrees with the results of the categorization algorithm using automated HFO markings.

50. Not clear that HFO rate is the best measure to assess clinical relevance of the HFO biomarker. Some interictal discharges occur more rarely than twice per minute, and yet may provide critically important clinical information, while normal phenomena may be accompanied by HFOs and occur more frequently. The paper then, can only conclude that use of HFO rates picked up by an agnostic detector alone may be unreliable - not that HFOs themselves are.

This is a very important point, which we do not address in this paper. HFO rate may not be the best indicator, especially if we do not know which HFOs “are normal and which are not”. However, HFO rate is far and away the standard of usage in HFO research. In fact, it is hard to find any papers at all that do not use rate as the standard. This is why we feel this paper is so important, as it shows that rate alone is not as reliable as advertised. It is critical to add in clinical context. We have made this latter point throughout the paper.

51. Qualitative statements that need to be supported with data & statistical testing:

Line 180: "quite close together"

Line 245: "qualitatively similar."

Line 251: "often extends across a wider region"

Lines 257-258: "very common for the answer to change from one night to the next"

Lines 268-9: "some patients...sufficient data were recorded....continued to have changes".

Line 286: "Other patients also had significant changes in HFO rate..."

Line 499: "agree with"

These are just some examples. The use of unsupported qualitative statements occurred throughout the text, detracting considerably from readability.

We have revised the entire paper to make it more quantitative, from the statistical evaluation of hypotheses to the addition of more patients and analysis to improve rigor. We hope this new approach has addressed this concern.

Minor:

52. The title and abstract should specify whether interictal or interictal + ictal HFOs are under consideration.... The reason that the prior studies selected NREM sleep is likely that interictal discharges are more frequent during this time, and because movement artifacts are minimized. As this topic is discussed in methods, this should be mentioned also.... Please provide references and justification for the frequency definition of ripples vs. fast ripples....

Done. Thank you for pointing out these issues.

53. The investigators speculate that the intracranial implantation itself causes HFO rates to vary over time. This is an intriguing hypothesis, but it is not testable with this dataset nor are prior literature cited to support it. It should therefore be presented as a speculation for future study, rather than established fact

With the new suggestion in #47, we have changed this part to refer to the known issues of moving seizure foci (Struck, et al.) and the implantation effect seen over months in the Neurovista data (Cook, et al.), which has better documentation.

Reviewers' comments:

Reviewer #1 (Remarks to the Author):

The authors have made substantial edits to the manuscript and additional analyses to the study. I only really have minor concerns (see below), but there is one thing that continues to bother me and one thing that I believe will prevent HFOs from being truly used universally in epilepsy centers: There is no large-scale comparison between the asymmetry measure (or any agreement measure) and surgical outcome. I would be highly intrigued if the authors could show that the patients that had less asymmetry were also patients with poor surgical outcomes and that the patients that had high asymmetry were patients with good or great outcomes. Until this study is performed, I do not think HFOs will move to all clinics.

Major Request:

From Table 2, it appears that you have the patient outcome for 80 Mayo clinic patients and 15 UM patients. For these same patients, can you please show the asymmetry distributions as a function of class (for each HFO detector)?

Minor:

With that being said, I think this is a nice large study (and fair) that brings to light the possible added value HFO analyses can provide to clinicians. Here are some minor concerns, which could go into the discussion of the paper.

- The authors make the argument that HFOs, if used properly and alongside with other imaging modalities and data, can help clinicians identify epileptic regions. In principle, I agree with this argument. However, I wonder if the authors could explain how clinicians should respond to discordance between HFO analyses and their visual inspection of intracranial EEG and other data?
- Why do you call the measure asymmetry where higher value is more symmetric? I would change the name if this is defined by authors.
- Fig. 2 caption : "The asymmetry was only computed for cases with at least on channel having" ... "on" should be "one"

Reviewer #2 (Remarks to the Author):

The authors have done significant work and addressed all of my concerns. I have some remaining minor comments.

line 141. Is "versus" the correct word here? Seems like "and" since both are shown?

line 150, Please state what are the "two distributions".

Figure 2. Q. How do we make sense of the peaks at -1 for Mayo using qHFO? Does this imply all HFOs are outside the SOZ, which would make the absence of HFO a perfect detector of SOZ? Is it something about the noise in these data, since this -1 peak is not present in UM cohort?

Figure 3, Please add day labels to Fig 3D, and make clear that lines 204-207 refer to Fig 3D.

lines 225,226, Please check this text with Figure 5, they appear inconsistent. Type 3 in text states days 7-8 and in figure the orange marks appear on day 6 and 7. Or maybe I misread the figure.

line 231, To improve readability consider adding a pointer to Figure 5 at the end of this sentence, something like "see Day 7, Time ? in Figure 5".

line 253, "(17±4)% had too few HFOs to analyze," Should this be clear in Fig 6A? Looks like sum of green bars in Mayo and UM is bigger than 17%.

line 263,264, "between any two cohorts for a given category," for the same detector, or across detectors too?

line 347, Please rephrase to remove double "not".

line 396, Maybe "highly" is too strong here.

That the two methods (qHFO and Hilbert) perform similarly is very nice, and important to this paper. But, in retrospect, partly undercuts the argument that the qHFO method is better. The authors might speculate why the Hilbert method does well in this study, even though that approach does not reject artifacts.

line 583, 597 remove bold.

Reviewer #3 (Remarks to the Author):

The paper has been revised extensively since the last submission, much improving its readability and focus. The manuscript is strengthened with addition of 8 patients (now total of 10) with long-term recordings & HFO analysis, and use of three published HFO detectors with comparison of results. The message of the paper has changed to emphasize the correlation of HFOs to the seizure onset zone in the aggregate data set, and then systematically explores the pitfalls of patient-specific HFO data. The conclusion of the paper (at least as I understood it) has also changed from concluding that HFOs are too unreliable for clinical utility to that HFOs are a useful epileptic biomarker but must be used with care - a very sensible take that is fully consistent with existing literature.

The major issue in the rewrite is that many of the paper's analysis methods have changed, and the conclusions changed accordingly. This is a serious problem for rigor/reproducibility as it is a form of cherry-picking. Instead of throwing out the previous max channel results, it would be better to present them (can go into supplementary information) as a way NOT to analyze HFOs, because the results are too unstable. This led to use of a new method that proved sufficiently robust (etc).

The remaining issues are all minor:

1. It's already well known that HFO/SOZ correlation exists, and stating this is simply a confirmation of previously published studies. The interesting point which is not stated is that the relationship is far from perfect, i.e. HFOs are commonly found outside the SOZ as is demonstrated in Fig 2 (most HFO asymmetry values are not close to 1). It would be helpful to provide a more nuanced conclusion, i.e. that there is a correlation but that HFOs often highlight regions outside the SOZ as well as (sub)regions within the SOZ - but not necessarily the entire SOZ. This is precisely why one would be interested in using them clinically, and it's also why HFO studies shifted from studying SOZ

correlation to assessing predictive value of HFO site resection for post-surgical seizure outcome.

2. The presentation of the analysis of temporal variability is much improved. This leads to a very appropriate discussion of how HFOs should be integrated into clinical practice. This probably deserves more attention. Specifically the point that HFOs cannot simply be used blindly to identify target brain sites should be emphasized throughout (starting in the introduction), and potential confounding issues such as unrecorded seizure types and spatial undersampling with the electrode placement should be mentioned as well. Conversely, while the HFO asymmetry index is well suited for the SOZ correlation analysis, it needs to be made clear that this approach is not a roadmap for clinical use of HFOs - to avoid confusion.

3. What does class a-d mean in Table 4? This table is completely uninterpretable without this info.

4. Intro is much too long, and could be tightened up to improve readability and make the salient points clearer.

5. Since there is currently very limited (if any) use of HFOs in clinical practice, instead of "modification in the strategy" ... (line 110) perhaps say "we use these results to suggest a strategy for clinical use of HFOs".

6. It would be helpful to define temporal variability at its first use in the paper.

7. line 376-377: "unknown ictal generation ... networks which also vary over time" - this is a surprising statement, as it would undermine the foundation of epilepsy surgery (in which seizure focus is considered to be at a fixed brain site that can then be surgically removed to control seizures). It is probably best to delete this, or alternatively cite literature to support it and provide a more complete discussion of the point that the authors are trying to make here.

We thank the reviewers for their helpful comments. We have addressed all reviewer comments, including the two “major requests” and have included five new plots in a supplement.

Reviewer #1

There is no large-scale comparison between the asymmetry measure (or any agreement measure) and surgical outcome. I would be highly intrigued if the authors could show that the patients that had less asymmetry were also patients with poor surgical outcomes and that the patients that had high asymmetry were patients with good or great outcomes. Until this study is performed, I do not think HFOs will move to all clinics. ... From Table 2, it appears that you have the patient outcome for 80 Mayo clinic patients and 15 UM patients. For these same patients, can you please show the asymmetry distributions as a function of class (for each HFO detector)?

We have added the analysis as Supplementary Figs. S1-2. However, the results do not show the ideal results that (we and) the reviewer wanted: in reality there is no difference between patients with good and bad outcome. We also added a similar analysis versus resected volume in S3-4. Again, there is no appreciable difference. We believe this is not a surprise for several reasons. First, asymmetry is definitely not a way to identify a region for resection. It is by definition retrospective, but more importantly it indiscriminantly groups all channels of one area (either SOZ or resected volume) and compares it with all channels outside of it. This forces many channels that might be in the “gray zone” to one side or the other. In practice, the SOZ is a clinical determination based on a great deal of data. The RV is based as much on anatomy as anything else. Thus both are only surrogates of the true epileptogenic zone. One can imagine many different combinations that will disrupt this analysis: a single frequently-spiking (with HFOs) channel outside the SOZ, a large number of quiet channels within the SOZ or RV that only activate with the seizure, etc. For this reason, we (nor anybody else) have never proposed asymmetry to be the method of using HFOs clinically. The asymmetry analysis alone is a coarse measurement: it mainly shows that HFOs are good, but not perfect—a common theme in many publications about HFOs. The advantage of the asymmetry analysis is that is simple and direct. The disadvantage is that it does not determine whether or how the information relates to clinical decisions. However, we believe this analysis shows quite clearly that the HFOs are highly correlated with epileptic tissue, which is the goal of this section of the paper. We are not trying to show how to use HFOs clinically in this work—we are trying to show that they are correlated with epileptic tissue, but that we need to be very careful with how they are used.

In short, we show that HFOs are intriguing, but agree that more work needs to be to develop the best protocols for incorporating HFOs into widespread clinical practice. The focus of this paper is not to address all the missing pieces, but to demonstrate that short amounts of time will not have enough information—future protocols should incorporate as many HFOs from as long as possible.

The first section of results now reads:

However, when stratified by patient outcome there was no clear relationship with asymmetry: results for patients with Class I outcome (Supplementary Fig. S1) were similar to those with Class II-V outcome (Suppl. Fig. S2) in both cohorts. The results were also similar

when comparing the resected volumes (Suppl. Figs. S3-4). Thus, asymmetry is not capable of identifying which patients had a good outcome: it is a nonspecific, retrospective measurement that compares regions rather than specific channels. Nevertheless, these results serve as a validation of HFOs as a potential biomarker. While some detectors have diminished performance in some cohorts, the HFOs are associated with epileptogenic tissue, regardless of detector. Furthermore, all HFO detectors show that the association of HFO rate with SOZ is statistically similar whether using just a 10-minute segments or longer segments to compute the HFO rate.

The discussion now states:

Our results corroborate with the findings of many other publications which used smaller data sets: while HFOs are associated with epileptogenic tissue, HFOs are not a perfect biomarker—HFOs occur outside of epileptogenic regions and even in patients without epilepsy.

Minor:

With that being said, I think this is a nice large study (and fair) that brings to light the possible added value HFO analyses can provide to clinicians. Here are some minor concerns, which could go into the discussion of the paper.

- The authors make the argument that HFOs, if used properly and alongside with other imaging modalities and data, can help clinicians identify epileptic regions. In principle, I agree with this argument. However, I wonder if the authors could explain how clinicians should respond to discordance between HFO analyses and their visual inspection of intracranial EEG and other data?*

This is an excellent point. One of the primary goals of our ongoing research is to answer this question. The challenge is that, to our knowledge, clinicians have never been given HFO data when making clinical decisions (save for the ongoing intraoperative HFO trial, which really is a different usage of HFOs). We hope to remedy that. In general, as we state in the paper, we think HFOs should be adjunctive to all other data: just as MRI, PET, SPECT, EEG, etc are sometimes discordant, HFOs will be as well. Clinical expertise is needed to reconcile all the data...but we need as much data as possible.

The final conclusion of the paper reads:

Deciding where to resect is an extremely complex process, and every piece of information is needed. Instead of using HFOs as an alternative to that process, we suggest that HFOs can serve as an important adjunct to current methods, providing unique, interictal information about the epileptic network and potentially increasing the clinician's confidence in final decisions.

- Why do you call the measure asymmetry where higher value is more symmetric? I would change the name if this is defined by authors.*

We apologize for the confusion. The asymmetry refers to how many of the HFOs are in one region (e.g. SOZ) versus another region (elsewhere). Our use is consistent with that used in

previous publications. The highest asymmetry values (+1) are when all HFOs are in the SOZ and no HFOs are outside. This is maximally asymmetric. The most symmetric case is when the mean HFO rate is the same within the SOZ and without the SOZ, in which case the asymmetry is zero.

• *Fig. 2 caption : “The asymmetry was only computed for cases with at least on channel having” ...”on” should be “one”*

Thank you for bringing this to our attention. We have made the correction.

Reviewer #2

line 141. Is “versus” the correct word here? Seems like “and” since both are shown?

Yes, “and” would be a more appropriate word choice. We have made the correction.

line 150, Please state what are the “two distributions”.

The two distributions are now noted.

Figure 2. Q. How do we make sense of the peaks at -1 for Mayo using qHFO? Does this imply all HFOs are outside the SOZ, which would make the absence of HFO a perfect detector of SOZ? Is it something about the noise in these data, since this -1 peak is not present in UM cohort?

We are sorry for not explaining this feature of the data more explicitly and have added text to the caption of Fig. 2 to help improve clarity on this point. The patients with asymmetries of -1 have no HFOs in the SOZ and some qHFO detections elsewhere. However, there may be other channels not in the SOZ that also do not have HFOs, and thus absence of HFOs is not specific to the SOZ even in those patients. You are correct that data quality and noise is the main contributor to this peak at -1. We have added this description to the text of the manuscript.

Figure 3, Please add day labels to Fig 3D, and make clear that lines 204-207 refer to Fig 3D.

We have added the day labels and clarification as requested.

lines 225,226, Please check this text with Figure 5, they appear inconsistent. Type 3 in text states days 7-8 and in figure the orange marks appear on day 6 and 7. Or maybe I misread the figure.

Thank you for noticing this. There was indeed a typo in the text.

line 231, To improve readability consider adding a pointer to Figure 5 at the end of this sentence, something like “see Day 7, Time ? in Figure 5”.

We have included this helpful suggestion.

line 253, “(17±4)% had too few HFOs to analyze,” Should this be clear in Fig 6A? Looks like sum of green bars in Mayo and UM is bigger than 17%.

This value is only clear from combining the data in Fig. 6A with the total number of subjects in each cohort that met the criteria for having enough HFOs to be analyze, though the correct procedure would be to perform a weighted average, not a direct sum. Just under 20% of the number of Mayo subjects plus about 10% of the number of UM NREM sleep subjects equals 17% of the total number of both subjects.

line 263,264, “between any two cohorts for a given category,” for the same detector, or across detectors too?

Thank you for pointing out the ambiguity. This value was for the same detector, as is not noted in the text.

line 347, Please rephrase to remove double “not”.

Done.

line 396, Maybe “highly” is too strong here.

We have removed this word.

That the two methods (qHFO and Hilbert) perform similarly is very nice, and important to this paper. But, in retrospect, partly undercuts the argument that the qHFO method is better. The authors might speculate why the Hilbert method does well in this study, even though that approach does not reject artifacts.

These two methods perform similarly to the extent that either method fully supports the conclusions of our paper—that HFOs are correlated with SOZ, but that 10 minutes of recording is not enough. However, Fig. 2 shows that the qHFO method has much better performance than the Hilbert method. On the Mayo data, which is known to have more noise and thus is the cohort more likely to show the impact of artifact rejection, the distribution of asymmetries for the Hilbert method look like a Gaussian centered about zero (Fig. 2C), whereas the qHFO detector clearly has a skew towards positive asymmetries (Fig. 2A). While more subtle, the asymmetries in Fig. 2D are better than those in Fig. 2F: Fig. 2D has only 1/15 with asymmetry below 0.2 and none below zero, whereas Fig. 2F has 6/15 below 0.2. We also note that while the qHFO detector includes the extra feature of artifact rejection, it also runs 20 times faster. The Hilbert method is likely too slow for clinical use, whereas the qHFO methods typically runs about 5 times faster than real time on a single core. We did not make a big deal about these differences because the purpose of the figure is to validate that the finding is not specific to our detector.

line 583, 597 remove bold.

Done.

Reviewer #3

The conclusion of the paper (at least as I understood it) has also changed from concluding that HFOs are too unreliable for clinical utility to that HFOs are a useful epileptic biomarker but must be used with care - a very sensible take that is fully consistent with existing literature.

We thank you for your supportive comments. The message and conclusion you state is exactly that which we were trying to get across in all versions of the manuscript. We apologize this was not clear in the early manuscript and are grateful for all the comments of the reviewers that have helped it become clear in the latest revision. However, we never stated nor implied that “HFOs are too unreliable for clinical utility”—our point has always been that the proposed method of using only 10 minutes of data, without any other analysis, is unreliable. However, based upon the comments of all reviewers, we now make it much more clear how HFOs should be used (see next comment).

The major issue in the rewrite is that many of the paper’s analysis methods have changed, and the conclusions changed accordingly. This is a serious problem for rigor/reproducibility as it is a form of cherry-picking. Instead of throwing out the previous max channel results, it would be better to present them (can go into supplementary information) as a way NOT to analyze HFOs, because the results are too unstable. This led to use of a new method that proved sufficiently robust (etc).

We would like to point out that our conclusion actually did not change—the only change is that we clarify that HFOs should be used in the context of all other data, and that more than 10 minutes of HFO data are needed. We completely agree on the importance of scientific rigor, reproducibility, and the need to avoid “cherry-picking.” We would like to reassure the reviewer that we have maintained these standards. In fact, we feel we have been completely fair, including all data from all patients—no matter how noisy—with automated analysis.

We have returned the max channel results to the supplement and methods in order to allay any concerns. Of note, this figure (S5) contains the same max channel data as before, but we have also included the results from the two new detectors that were added to Revision 1. We point out this figure in the text, stating:

These results are similar to previous work showing that simply choosing the channel with maximum HFOs is not always reliable for identifying the SOZ (25).

The remaining issues are all minor:

1. It's already well known that HFO/SOZ correlation exists, and stating this is simply a confirmation of previously published studies. The interesting point which is not stated is that the relationship is far from perfect, i.e. HFOs are commonly found outside the SOZ as is demonstrated in Fig 2 (most HFO asymmetry values are not close to 1). It would be helpful to provide a more nuanced conclusion, i.e. that there is a correlation but that HFOs often highlight

regions outside the SOZ as well as (sub)regions within the SOZ - but not necessarily the entire SOZ. This is precisely why one would be interested in using them clinically, and it's also why HFO studies shifted from studying SOZ correlation to assessing predictive value of HFO site resection for post-surgical seizure outcome.

We agree. We have added text to talk about this imperfect relationship, and the need to exercise considerable clinical correlation interpreting HFOs. We specifically point out in our paper that HFOs were in fact first discovered in normal brain activity.

2. The presentation of the analysis of temporal variability is much improved. This leads to a very appropriate discussion of how HFOs should be integrated into clinical practice. This probably deserves more attention. Specifically the point that HFOs cannot simply be used blindly to identify target brain sites should be emphasized throughout (starting in the introduction), and potential confounding issues such as unrecorded seizure types and spatial undersampling with the electrode placement should be mentioned as well. Conversely, while the HFO asymmetry index is well suited for the SOZ correlation analysis, it needs to be made clear that this approach is not a roadmap for clinical use of HFOs - to avoid confusion.

We agree and have made several changes throughout the text to talk about these points.

Unfortunately, it is still not clear how best to use HFOs in clinical practice. We hope that future work will start looking at clinical implementation.

3. What does class a-d mean in Table 4? This table is completely uninterpretable without this info.

Thank you for pointing out this missing information. They correspond to the categorization classes described in the text and in Fig. 6. We have adjusted the caption of Table 4 to make this clear.

4. Intro is much too long, and could be tightened up to improve readability and make the salient points clearer.

Given the broad audience of Nature Communications, the request of this and other reviewers to add even more explanation to the intro, the mildly-contrarian nature of this paper (necessitating a description of the current state of the field), and the general format of NatComm that puts methods after results, we feel it is necessary to leave the intro as it is.

5. Since there is currently very limited (if any) use of HFOs in clinical practice, instead of "modification in the strategy" ... (line 110) perhaps say "we use these results to suggest a strategy for clinical use of HFOs".

This is a valid point. This sentence has been modified to be more specific and accurate.

6. It would be helpful to define temporal variability at its first use in the paper.

Done.

7. line 376-377: "unknown ictal generation ... networks which also vary over time" - this is a surprising statement, as it would undermine the foundation of epilepsy surgery (in which seizure focus is considered to be at a fixed brain site that can then be surgically removed to control seizures). It is probably best to delete this, or alternatively cite literature to support it and provide a more complete discussion of the point that the authors are trying to make here.

We apologize for the confusion and have added text to make this point more clear. The text now reads:

It is possible that the HFO clusters move not because they are an unreliable biomarker of ictal activity, but because they are tightly coupled to underlying, unknown ictal generation and propagation networks, which also vary over time. The prime example is multifocal epilepsy, which has multiple seizure types arising from more than one region of the brain: such patients require special care to assure enough time has been allotted to capture the full extent of the seizure network(s). Even in patients with a single epileptic focus, each individual seizure may arise and/or involve from a sub-section of that focus. Thus, there appears to be a higher level of spatial and temporal resolution to ictal networks than the broad, binary categorization of tissue being epileptogenic or not. Unfortunately, accessing these underlying propagation networks is beyond current technological capabilities, though high resolution microwires (39) and microarrays (40) have found evidence of such effects. This situation further illustrates the need to incorporate as much clinical data—and recording time—as necessary when making surgical decisions.

REVIEWERS' COMMENTS:

Reviewer #1 (Remarks to the Author):

I believe that the authors made substantial improvement to their study and recommend for publication.

Reviewer #2 (Remarks to the Author):

Thank you for addressing my last round of comments. I have no further comments, and recommend the manuscript now be accepted.

Reviewer #3 (Remarks to the Author):

The authors have satisfactorily addressed my main critiques. Just one minor suggestion: The text is written as if the SOZ were the gold standard for id'ing epileptogenic cortex. As the authors well know, it is an imperfect indicator (if it were perfect there would be no need to investigate new biomarkers such as HFOs), as evidenced by its imperfect prediction of surgical outcome (Weiss et al, Neurology 2015 provides this data).

I suggest making it clearer in the text (apart from the late/brief discussion of multiple sources). Specifically, make it clear that purpose of the asymmetry index is to provide evidence that the detected HFOs are clinically relevant. But, the presence of HFOs outside the SOZ is not a negative - rather it should highlight areas that clinicians should consider including in the planned resection, depending on how these areas relate to other known factors e.g. EEG spread, clinical semiology, MRI abnormalities etc. Similarly, if only a portion of the SOZ can be resected (again, see report on this in Weiss et al 2015), targeting the HFO+ sites within the SOZ (as was done in Modur et al, 2011) may optimize outcomes.

Response to reviewer 3:

The authors have satisfactorily addressed my main critiques. Just one minor suggestion: The text is written as if the SOZ were the gold standard for id'ing epileptogenic cortex. As the authors well know, it is an imperfect indicator (if it were perfect there would be no need to investigate new biomarkers such as HFOs), as evidenced by its imperfect prediction of surgical outcome (Weiss et al, Neurology 2015 provides this data).

I suggest making it clearer in the text (apart from the late/brief discussion of multiple sources). Specifically, make it clear that purpose of the asymmetry index is to provide evidence that the detected HFOs are clinically relevant. But, the presence of HFOs outside the SOZ is not a negative - rather it should highlight areas that clinicians should consider including in the planned resection, depending on how these areas relate to other known factors e.g. EEG spread, clinical semiology, MRI abnormalities etc. Similarly, if only a portion of the SOZ can be resected (again, see report on this in Weiss et al 2015), targeting the HFO+ sites within the SOZ (as was done in Modur et al, 2011) may optimize outcomes.

We have added the following changes to include the reviewer's suggestion:

Introduction:

The SOZ is only an estimation of the true epileptogenic zone: it is the best current standard but removing it does not always lead to seizure freedom, necessitating the search for additional biomarkers (2, 3). *(which are the Weiss and Modur citations as suggested)*

Discussion:

We note that the asymmetry provides evidence that the HFOs are clinically relevant, but is retrospective and cannot be used for prospective determination of SOZ.

Conclusion:

The presence of HFOs outside the SOZ can represent either 'normal' HFOs or areas that should be considered as part of the epileptic network. As there is currently no perfect method to determine whether an HFO is epileptic (7), clinical decisions should not depend solely upon the presence of HFOs, but should also include all other clinical data.